# Adversarial Robustness with Semi-Infinite Constrained Learning

**Alexander Robey**[*]
University of Pennsylvania
arobey1@seas.upenn.edu

**Luiz F. O. Chamon**[*]
University of California, Berkeley
lfochamon@berkeley.edu

**George J. Pappas**
University of Pennsylvania
pappasg@seas.upenn.edu

**Hamed Hassani**
University of Pennsylvania
hassani@seas.upenn.edu

**Alejandro Ribeiro**
University of Pennsylvania
aribeiro@seas.upenn.edu

## Abstract

Despite strong performance in numerous applications, the fragility of deep learning to input perturbations has raised serious questions about its use in safety-critical domains. While adversarial training can mitigate this issue in practice, state-of-the-art methods are increasingly application-dependent, heuristic in nature, and suffer from fundamental trade-offs between nominal performance and robustness. Moreover, the problem of finding worst-case perturbations is non-convex and underparameterized, both of which engender a non-favorable optimization landscape. Thus, there is a gap between the theory and practice of adversarial training, particularly with respect to *when* and *why* adversarial training works. In this paper, we take a constrained learning approach to address these questions and to provide a theoretical foundation for robust learning. In particular, we leverage semi-infinite optimization and non-convex duality theory to show that adversarial training is equivalent to a statistical problem over perturbation distributions, which we characterize completely. Notably, we show that a myriad of previous robust training techniques can be recovered for particular, sub-optimal choices of these distributions. Using these insights, we then propose a hybrid Langevin Monte Carlo approach of which several common algorithms (e.g., PGD) are special cases. Finally, we show that our approach can mitigate the trade-off between nominal and robust performance, yielding state-of-the-art results on MNIST and CIFAR-10. Our code is available at: https://github.com/arobey1/advbench.

## 1 Introduction

Learning is at the core of many modern information systems, with wide-ranging applications in clinical research [1–4], smart grids [5–7], and robotics [8–10]. However, it has become clear that learning-based solutions suffer from a critical lack of robustness [11–17], leading to models that are vulnerable to malicious tampering and unsafe behavior [18–22]. While robustness has been studied in statistics for decades [23–25], this issue has been exacerbated by the opacity, scale, and non-convexity of modern learning models, such as convolutional neural network (CNNs). Indeed, the pernicious nature of these vulnerabilities has led to a rapidly-growing interest in improving the so-called *adversarial robustness* of modern ML models. To this end, a great deal of empirical evidence has shown *adversarial training* to be the most effective way to obtain robust classifiers, wherein models are trained on perturbed samples rather than directly on clean data [26–31]. While this approach is now ubiquitous in practice, adversarial training faces two fundamental challenges.

---

[*] Alexander Robey and Luiz F. O. Chamon contributed equally to this work.

35th Conference on Neural Information Processing Systems (NeurIPS 2021).

Firstly, it is well-known that obtaining *worst-case*, adversarial perturbations of data is challenging in the context of deep neural networks (DNNs) [32, 33]. While gradient-based methods have been shown to be empirically effective at finding perturbations that lead to misclassification, there are no guarantees that these perturbations are truly worst-case due to the non-convexity of most commonly-used ML function classes [34]. Moreover, whereas optimizing the parameters of a DNNs is typically an overparameterized problem, finding worst-case perturbations is severely underparametrized and therefore does not enjoy the benign optimization landscape of standard training [35–39]. For this reason, state-of-the-art adversarial attacks increasingly rely on heuristics such as random initializations, multiple restarts, pruning, and other *ad hoc* training procedures [40–49].

The second challenge faced by adversarial training is that it engenders a fundamental trade-off between robustness and nominal performance [50–52]. In practice, penalty-based methods that incorporate clean data into the training objective are often used to overcome this issue [53–56]. However, while empirically successful, these methods cannot typically guarantee nominal or adversarial performance outside of the training samples. Indeed, classical learning theory [57, 58] provides generalization bounds only for the aggregated objective and not each individual penalty term. Additionally, the choice of the penalty parameter is not straightforward and depends on the underlying learning task, making it difficult to transfer across applications and highly dependent on domain expert knowledge.

**Contributions.** To summarize, there is a significant gap between the theory and practice of robust learning, particularly with respect to *when* and *why* adversarial training works. In this paper, we study the algorithmic foundations of robust learning toward understanding the fundamental limits of adversarial training. To do so, we leverage semi-infinite constrained learning theory, providing a theoretical foundation for gradient-based attacks and mitigating the issue of nominal performance degradation. In particular, our contributions are as follows:

- We show that adversarial training is equivalent to a stochastic optimization problem over a specific, *non-atomic* distribution, which we characterize using recent non-convex duality results [59, 60]. Further, we show that a myriad of previous adversarial attacks reduce to particular, sub-optimal choices of this distribution.
- We propose an algorithm to solve this problem based on stochastic optimization and Markov chain Monte Carlo. Gradient-based methods can be seen as limiting cases of this procedure.
- We show that our algorithm outperforms state-of-the-art baselines on MNIST and CIFAR-10. In particular, our approach yields a ResNet-18 classifiers which simultaneously achieves greater than 50% adversarial accuracy and greater than 85% clean accuracy on CIFAR-10, which represents a significant improvement over the previous state-of-the-art.
- We provide generalization guarantees for the empirical version of this algorithm, showing how to effectively limit the nominal performance degradation of robust classifiers.

## 2 Problem formulation

Throughout this paper, we consider a standard classification setting in which the data is distributed according to an unknown joint distribution $\mathcal{D}$ over instance-label pairs $(\mathbf{x}, y)$. In this setting, the instances $\mathbf{x} \in \mathcal{X}$ are assumed to be supported on a compact subset of $\mathbb{R}^d$, and each label $y \in \mathcal{Y} := \{1, \dots, K\}$ denotes the class of a given instance $\mathbf{x}$. By $(\Omega, \mathcal{B})$ we denote the underlying measurable space for this setting, where $\Omega = \mathcal{X} \times \mathcal{Y}$ and $\mathcal{B}$ denotes its Borel $\sigma$-algebra. Furthermore, we assume that the joint distribution $\mathcal{D}$ admits a density $\mathfrak{p}(\mathbf{x}, y)$ defined over the sets of $\mathcal{B}$.

At a high level, our goal is to learn a classifier which can correctly predict the label $y$ of a corresponding instance $\mathbf{x}$. To this end, we let $\mathcal{H}$ be a hypothesis class containing functions $f_{\boldsymbol{\theta}} : \mathbb{R}^d \to \mathcal{S}^K$ parameterized by vectors $\boldsymbol{\theta} \in \Theta \subset \mathbb{R}^p$, where we assume that the parameter space $\Theta$ is compact and by $\mathcal{S}^K$ we denote the $(K-1)$-simplex. We also assume that $f_{\boldsymbol{\theta}}(\mathbf{x})$ is differentiable with respect to $\boldsymbol{\theta}$ and $\mathbf{x}$.[1] To make a prediction $\hat{y} \in \mathcal{Y}$, we assume that the simplex $\mathcal{S}^K$ is mapped to the set of classes $\mathcal{Y}$ via $\hat{y} \in \operatorname{argmax}_{k \in \mathcal{Y}} [f_{\boldsymbol{\theta}}(\mathbf{x})]_k$ with ties broken arbitrarily. In this way, we can think of the $k$-th output of the classifier as representing the probability that $y = k$. Given this notation, the statistical problem of learning a classifier that accurately predicts the label $y$ of a given instance $\mathbf{x}$

---

[1]Note that the classes of support vector machines, logistic classifiers, and convolutional neural networks (CNNs) with softmax outputs can all be described by this formalism.

drawn randomly from $\mathcal{D}$ can be formulated as follows:

$$\min_{\boldsymbol{\theta} \in \Theta} \mathbb{E}_{(\mathbf{x},y) \sim \mathcal{D}} \left[ \ell\big(f_{\boldsymbol{\theta}}(\mathbf{x}), y\big) \right]. \tag{P-NOM}$$

Here $\ell$ is a $[0, B]$-valued loss function and $\ell(\cdot, y)$ is $M$-Lipschitz continuous for all $y \in \mathcal{Y}$. We assume that $(\mathbf{x}, y) \mapsto \ell\big(f_{\boldsymbol{\theta}}(\mathbf{x}), y\big)$ is integrable so that the objective in (P-NOM) is well-defined; we further assume that this map is an element of the Lebgesgue space $L^p(\Omega, \mathcal{B}, \mathfrak{p})$ for some fixed $p \in (1, \infty)$.

**Formulating the robust training objective.** For common choices of the hypothesis class $\mathcal{H}$, including DNNs, classifiers obtained by solving (P-NOM) are known to be sensitive to small, norm-bounded input perturbations [61]. In other words, it is often straightforward to find a relatively small perturbations $\boldsymbol{\delta}$ such that the classifier correctly predicts the label $y$ of $\mathbf{x}$, but misclassifies the perturbed sample $\mathbf{x} + \boldsymbol{\delta}$. This has led to increased interest in the robust analog of (P-NOM), namely,

$$P_{\mathrm{R}}^{\star} \triangleq \min_{\boldsymbol{\theta} \in \Theta} \mathbb{E}_{(\mathbf{x},y) \sim \mathcal{D}} \left[ \max_{\boldsymbol{\delta} \in \Delta} \ell\big(f_{\boldsymbol{\theta}}(\mathbf{x} + \boldsymbol{\delta}), y\big) \right]. \tag{P-RO}$$

In this optimization program, the set $\Delta \subset \mathbb{R}^d$ denotes the set of valid perturbations[2]. Typically, $\Delta$ is chosen to be a ball with respect to a given metric on Euclidean space, i.e., $\Delta = \{\boldsymbol{\delta} \in \mathbb{R}^d : \|\boldsymbol{\delta}\| \leq \epsilon\}$. However, in this paper we make no particular assumption on the specific form of $\Delta$. In particular, our results apply to arbitrary perturbation sets, such as those used in [62–66].

Analyzing conditions under which (P-RO) can be (probably approximately) solved from data remains an active area of research. While bounds on the Rademacher complexity [67, 68] and VC dimension [67–71] of the robust loss

$$\ell_{\mathrm{adv}}(f_{\boldsymbol{\theta}}(\mathbf{x}), y) = \max_{\boldsymbol{\delta} \in \Delta} \ell\big(f_{\boldsymbol{\theta}}(\mathbf{x} + \boldsymbol{\delta}), y\big) \tag{1}$$

have been derived for an array of losses $\ell$ and hypothesis classes $\mathcal{H}$, there are still open questions on the effectiveness and sample complexity of adversarial learning [68]. Moreover, because in general the map $\boldsymbol{\delta} \mapsto \ell\big(f_{\boldsymbol{\theta}}(\mathbf{x} + \boldsymbol{\delta}), y\big)$ is non-concave, evaluating the maximum in (1) is not straightforward. To this end, the most empirically effective strategy for approximating the robust loss is to leverage the differentiability of modern ML models with respect to their inputs. More specifically, by computing gradients of such models, one can approximate the value of (1) using projected gradient ascent. For instance, in [27, 72] an perturbation $\boldsymbol{\delta}$ is computed for a fixed parameter $\boldsymbol{\theta}$ and data point $(\mathbf{x}, y)$ by repeatedly applying

$$\boldsymbol{\delta} \leftarrow \textstyle\prod_{\Delta} \Big[ \boldsymbol{\delta} + \eta \, \mathrm{sign} \big[ \nabla_{\boldsymbol{\delta}} \, \ell\big(f_{\boldsymbol{\theta}}(\mathbf{x} + \boldsymbol{\delta}), y\big) \big] \Big], \tag{2}$$

where $\prod_{\Delta}$ denotes the projection onto $\Delta$ and $\eta > 0$ is a fixed step size. This idea is at the heart of adversarial training, in which (2) is iteratively applied to approximately evaluate the robust risk in the objective of (P-RO); the parameters $\boldsymbol{\theta}$ can then be optimized with respect to this robust risk.

**Common pitfalls for adversarial training.** Their empirical success notwithstanding, gradient-based approaches to adversarial training are not without drawbacks. One pitfall is the fact that gradient-based algorithms are not guaranteed to provide optimal (or even near-optimal) perturbations, since $\mathbf{x} \mapsto \ell(f_{\boldsymbol{\theta}}(\mathbf{x}), y)$ is typically not a concave function. Because of this, heuristics [73, 74] are often needed to improve the solutions obtained from (2). Furthermore, adversarial training often degrades the performance of the model on clean data [52, 75, 76]. In practice, penalty-based approaches are used to empirically overcome this issue [53], although results are not guaranteed to generalize outside of the training sample. Indeed, classical learning theory guarantees generalization in terms of the aggregated objective and not in terms of the robustness requirements it may describe [57, 58, 60].

In the remainder of this paper, we address these pitfalls by leveraging semi-infinite constrained learning theory. To do so, we explicitly formulate the problem of finding the most robust classifier among those that have good nominal performance. Next, we show that (P-RO) is equivalent to a stochastic optimization problem that can be related to numerous adversarial training methods (Section 3). We then provide generalization guarantees for the constrained robust learning problem when solve using empirical (*unconstrained*) risk minimization (Section 4). Finally, we derive an algorithm based on a Langevin MCMC sampler of which (2) is a particular case (Section 5).

---

[2]Note that $f_{\boldsymbol{\theta}}$ must now be defined on $\mathcal{X} \oplus \Delta$, where $\oplus$ denotes the Minkowski (set) sum. In a slight abuse of notation, we will refer to this set as $\mathcal{X}$ from now on.

# 3 Dual robust learning

In the previous section, we argued that while empirically successful, adversarial training is not without shortcomings. In this section, we develop the theoretical foundations needed to tackle the two challenges of (P-RO): (a) finding worst-case perturbations, i.e., evaluating the robust loss defined in (1) and (b) mitigating the trade-off between robustness and nominal performance. To address these challenges, we first propose the following constrained optimization problem which explicitly captures the trade-off between robustness and nominal performance:

$$
\begin{aligned}
P^\star \triangleq \min_{\boldsymbol{\theta} \in \Theta} \quad & \mathbb{E}_{(\mathbf{x},y) \sim \mathcal{D}} \left[ \max_{\boldsymbol{\delta} \in \Delta} \ell\big(f_{\boldsymbol{\theta}}(\mathbf{x} + \boldsymbol{\delta}), y\big) \right] \\
\text{subject to} \quad & \mathbb{E}_{(\mathbf{x},y) \sim \mathcal{D}} \left[ \ell\big(f_{\boldsymbol{\theta}}(\mathbf{x}), y\big) \right] \leq \rho
\end{aligned}
\tag{P-CON}
$$

where $\rho \geq 0$ is a desired nominal performance level. At a high level, (P-CON) seeks the most robust classifier $f_{\boldsymbol{\theta}}(\cdot)$ among those classifiers that have strong nominal performance. In this way, (P-CON) is directly designed to address the trade-off between robustness and accuracy, and as such (P-CON) will be the central object of study in this paper. We note that at face value, the statistical constraint in (P-CON) is challenging to enforce in practice, especially given the well-known difficulty in solving the unconstrained analog (P-RO). Following [60, 77], our approach is to use duality to obtain solutions for (P-CON) that generalize with respect to both adversarial and nominal performance.

**Computing worst-case perturbations.** Before tackling the constrained problem (P-CON), we begin by consider its unconstrained version, namely, (P-RO). We start by rewriting (P-RO) using an epigraph formulation of the maximum function to obtain the following semi-infinite program:

$$
\begin{aligned}
P_{\mathrm{R}}^\star = \min_{\boldsymbol{\theta} \in \Theta, \, t \in L^p} \quad & \mathbb{E}_{(\mathbf{x},y) \sim \mathcal{D}} \big[ t(\mathbf{x}, y) \big] \\
\text{subject to} \quad & \ell\big(f_{\boldsymbol{\theta}}(\mathbf{x} + \boldsymbol{\delta}), y\big) \leq t(\mathbf{x}, y), \quad \text{for almost every } (\mathbf{x}, \boldsymbol{\delta}, y) \in \mathcal{X} \times \Delta \times \mathcal{Y}.
\end{aligned}
\tag{PI}
$$

Note that (PI) is indeed equivalent to (P-RO) since

$$
\max_{\boldsymbol{\delta} \in \Delta} \ell\big(f_{\boldsymbol{\theta}}(\mathbf{x} + \boldsymbol{\delta}), y\big) \leq t(\mathbf{x}, y) \iff \ell\big(f_{\boldsymbol{\theta}}(\mathbf{x} + \boldsymbol{\delta}), y\big) \leq t(\mathbf{x}, y) \text{ for all } \boldsymbol{\delta} \in \Delta.
\tag{3}
$$

While at first it may seem that we have made (P-CON) more challenging to solve by transforming an unconstrained problem into an infinitely-constrained problem, notice that (PI) is no longer a composite minimax problem. Furthermore, it is *linear* in $t$, indicating that (PI) should be amenable to approaches based on Lagrangian duality. Indeed, the following proposition shows that (PI) can be used to obtain a statistical counterpart of (P-RO).

**Proposition 3.1.** *If* $(\mathbf{x}, y) \mapsto \ell\big(f_{\boldsymbol{\theta}}(\mathbf{x}), y\big) \in L^p$ *for* $p \in (1, \infty)$*, then* (P-RO) *can be written as*

$$
P_{\mathrm{R}}^\star = \min_{\boldsymbol{\theta} \in \Theta} \, p(\boldsymbol{\theta}),
\tag{PII}
$$

*for the primal function*

$$
p(\boldsymbol{\theta}) \triangleq \max_{\lambda \in \mathcal{P}^q} \, \mathbb{E}_{(\mathbf{x},y) \sim \mathcal{D}} \left[ \mathbb{E}_{\boldsymbol{\delta} \sim \lambda(\boldsymbol{\delta} | \mathbf{x}, y)} [\ell(f_{\boldsymbol{\theta}}(\mathbf{x} + \boldsymbol{\delta}), y)] \right],
\tag{4}
$$

*where* $\mathcal{P}^q$*, with* $\frac{1}{p} + \frac{1}{q} = 1$*, is the subspace of* $L^q$ *containing almost everywhere non-negative functions such that* $\mathfrak{p}(\mathbf{x}, y) = 0 \Rightarrow \lambda(\boldsymbol{\delta} \mid \mathbf{x}, y) = 0$ *and* $\int \lambda(\boldsymbol{\delta} \mid \mathbf{x}, y) d\boldsymbol{\delta} = 1$ *for almost every* $(\mathbf{x}, y) \in \mathcal{X} \times \mathcal{Y}$*.*

The proof is provided in Appendix B. Informally, Proposition 3.1 shows that the robust learning problem in (P-RO) can be recast as a problem of optimizing over a set of probability distributions $\mathcal{P}^q$ taking support over $\Delta$. This establishes an equivalence between the traditional robust learning problem (P-RO), where the maximum is taken over perturbations $\boldsymbol{\delta} \in \Delta$ of the input, and its stochastic version (PII), where the maximum is taken over a conditional distribution over perturbations $\boldsymbol{\delta} \sim \lambda(\boldsymbol{\delta}|\mathbf{x}, y)$. Notably, a variety of training formulations can be seen as special cases of (PII). In fact, for particular sub-optimal choices of this distribution, paradigms such as random data augmentation and distributionally robust optimization can be recovered (see Appendix A for details).

As we remarked in Section 2, for many modern function classes, the task of evaluating the adversarial loss (1) is a nonconcave optimization problem, which is challenging to solve in general. Thus,

Proposition 3.1 can be seen as lifting the nonconcave inner problem $\max_{\delta \in \Delta} \ell(f_{\boldsymbol{\theta}}(\mathbf{x}), y)$ to the equivalent *linear* optimization problem in (4) over probability distributions $\lambda \in \mathcal{P}^q$. This dichotomy parallels the one that arises in PAC vs. agnostic PAC learning. Indeed, while the former seeks a deterministic map $(\boldsymbol{\theta}, \mathbf{x}, y) \mapsto \boldsymbol{\delta}$, the latter considers instead a distribution of perturbations over $\boldsymbol{\delta}|\mathbf{x}, y$ parametrized by $\boldsymbol{\theta}$. In fact, since (PII) is obtained from (P-RO) through semi-infinite duality, the density of this distribution is exactly characterized by the dual variables $\lambda$.

Note that while (PII) was obtained using Lagrangian duality, it can also be seen as a linear lifting of the maximization in (1). From this perspective, while recovering (1) would require $\lambda$ to be atomic, Proposition 3.1 shows that this is not necessary as long as $\ell(f_{\boldsymbol{\theta}}(\mathbf{x}), y)$ is an element of $L^p$. That is, because $\mathcal{P}^q$ does not contain any Dirac distributions, the optimal distribution $\lambda^\star$ for the maximization problem in (4) is *non-atomic*.[3] Hence, Proposition 3.1 does not show that (PII) finds worst-case perturbations that achieve the maximum in the objective of (P-RO). It does, however, show that finding worst-case perturbation is not essential to find a solution of (P-RO)

**Exact solutions for the maximization in** (PII). While (PII) provides a new constrained formulation for (P-RO), the objectives of both (PII) and (P-RO) still involve the solution of a non-trivial maximization. However, whereas the maximization problem in (P-RO) is a finite-dimensional problem which is nonconcave for most modern function classes, the maximization in (PII) is a linear, variational problem regardless of the function class. We can therefore leverage variational duality theory to obtain a full characterization of the optimal distribution $\lambda^\star$ when $p = 2$.

**Proposition 3.2** (Optimal distribution for (PII)). *Let $p = 2$ (and $q = 2$) in Proposition 3.1 and let $[z]_+ = \max(0, z)$. For each $(\mathbf{x}, y) \in \Omega$, there exists constants $\gamma(\mathbf{x}, y) > 0$ and $\mu(\mathbf{x}, y) \in \mathbb{R}$ s.t.*

$$\lambda^\star(\boldsymbol{\delta}|\mathbf{x}, y) = \left[ \frac{\ell(f_{\boldsymbol{\theta}}(\mathbf{x} + \boldsymbol{\delta}), y) - \mu(\mathbf{x}, y)}{\gamma(\mathbf{x}, y)} \right]_+ \tag{5}$$

*is a solution of the maximization in (4). In particular, the value of $\mu(\mathbf{x}, y)$ is such that*

$$\int_\Delta \left[ \ell(f_{\boldsymbol{\theta}}(\mathbf{x} + \boldsymbol{\delta}), y) - \mu(\mathbf{x}, y) \right]_+ d\boldsymbol{\delta} = \gamma(\mathbf{x}, y) \quad \forall (\mathbf{x}, y) \in \mathcal{X} \times \mathcal{Y}. \tag{6}$$

The proof is provided in Appendix C. This proposition shows that when $(\mathbf{x}, y) \mapsto \ell\big(f_{\boldsymbol{\theta}}(\mathbf{x}), y\big) \in L^2$, we can obtain a closed-form expression for the distribution $\lambda^\star$ that maximizes the objective of (PII). Moreover, this distribution is proportional to a truncated version of the loss of the classifier. Note that the assumption that the loss belongs to $L^2$ is mild given that the compactness of $\mathcal{X}$, $\mathcal{Y}$, and $\Delta$ imply that $L^{p_1} \subset L^{p_2}$ for $p_1 > p_2$. It is, however, fundamental to obtain the closed-form solution in Proposition 3.2 since it allows (4) to be formulated as a strongly convex constrained problem whose primal solution (5) can be recovered from its dual variables (namely, $\gamma$ and $\mu$). To illustrate this result, we consider two particular *suboptimal* choices for the constants $\mu$ and $\gamma$.

**Special case I: over-smoothed $\lambda^\star$.** Consider the case when $\gamma(\mathbf{x}, y)$ is taken to be the normalizing constant $\int_\Delta \ell(f_{\boldsymbol{\theta}}(\mathbf{x} + \boldsymbol{\delta}), y) d\boldsymbol{\delta}$ for each $(\mathbf{x}, y) \in \Omega$. As the loss function $\ell$ is non-negative, (6) implies that $\mu(\mathbf{x}, y) = 0$, and the distribution defined in (5) can be written as

$$\lambda(\boldsymbol{\delta}|\mathbf{x}, y) = \frac{\ell(f_{\boldsymbol{\theta}}(\mathbf{x} + \boldsymbol{\delta}), y)}{\int_\Delta \ell(f_{\boldsymbol{\theta}}(\mathbf{x} + \boldsymbol{\delta}), y) d\boldsymbol{\delta}}, \tag{7}$$

meaning that $\lambda(\boldsymbol{\delta}|\mathbf{x}, y)$ is exactly proportional to the loss $\ell(f_{\boldsymbol{\theta}}(\mathbf{x} + \boldsymbol{\delta}), y)$ on a perturbed copy of the data. Thus, for this choice of $\gamma$ and $\mu$, the distribution $\lambda$ in (7) is an over-smoothed version of the optimal distribution $\lambda^\star$. In our experiments, we will use this over-smoothed approximation of $\lambda^\star$ to derive an MCMC-style sampler, which yields state-of-the-art performance on standard benchmarks.

**Special case II: under-smoothed $\lambda^\star$.** It is also of interest to consider the case in which $\gamma$ approaches zero. In the proof of Proposition 3.2, we show that the value of $\mu$ is fully determined by $\gamma$ and that $\gamma$ is directly related to the smoothness of the optimal distribution; in fact, $\gamma$ is equivalent to a bound on the $L^2$ norm of $\lambda^\star$. In this way, as we take $\gamma$ to zero, we find that $\mu$ approaches $\max_{\boldsymbol{\delta} \in \Delta} \ell(f_{\boldsymbol{\theta}}(\mathbf{x} + \boldsymbol{\delta}), y)$, meaning that the distribution is truncated so that mass is only placed on those perturbations $\boldsymbol{\delta}$ which induce the maximum loss. Thus, in the limit, $\lambda$ approaches an atomic distribution concentrated

---

[3]Proposition 3.1 does not account for $p \in \{1, \infty\}$ for conciseness. Nevertheless, neither of the dual spaces $L^{1^*}$ or $L^{\infty^*}$ contain Dirac distributions, meaning that for $p \in \{1, \infty\}$, $\lambda^\star$ would remain non-atomic.

entirely at a perturbation $\delta^\star$ that maximizes the loss. Interestingly, this is the same distribution that would be needed to recover the solution to the inner maximization as in (P-RO). This highlights the fact that although recovering the optimal $\delta^\star$ in (P-CON) would require $\lambda^\star$ to be atomic, the condition that $\gamma > 0$ means that $\lambda^\star$ need not be atomic.

These two cases illustrate the fundamental difference between (P-RO) and (PII): Whereas in (P-RO) we search for worst-case perturbations, in (PII) we seek a method to sample perturbations $\delta$ from the perturbation distribution $\lambda^\star$. Thus, given a method for sampling $\delta \sim \lambda^\star(\delta|\mathbf{x}, y)$, the max in (P-RO) can be replaced by an expectation, allowing us to consider the following optimization problem:

$$P_R^\star = \min_{\boldsymbol{\theta} \in \Theta} \mathbb{E}_{(\mathbf{x}, y) \sim \mathcal{D}} \left[ \mathbb{E}_{\boldsymbol{\delta} \sim \lambda^\star(\boldsymbol{\delta}|\mathbf{x}, y)} \left[ \ell(f_{\boldsymbol{\theta}}(\mathbf{x} + \boldsymbol{\delta}, y)) \right] \right]. \tag{PIII}$$

Notice that crucially this problem is non-composite in the sense that it no longer contains an inner maximization. To this end, in Section 5, we propose a scheme that can be used to sample from a close approximation of $\lambda^\star$ toward evaluating the inner expectation in (PIII).

## 4   Solving the constrained learning problem

So far, we have argued that (P-CON) captures the problem of finding the most robust model with high nominal performance and we have shown the the minimax objective of (P-CON) can be rewritten as a stochastic optimization problem over perturbation distributions. In this section, we address the distinct yet related issue of satisfying the constraint in (P-CON), which is a challenging task in practice given the statistical and potentially non-convex nature of the problem. Further complicating matters is the fact that by assumption we have access to the data distribution $\mathcal{D}$ only through samples $(\mathbf{x}, y) \sim \mathcal{D}$, which means that in practice we cannot evaluate either of the expectations in (P-CON). To overcome these obstacles, given a dataset $\{(\mathbf{x}_n, y_n)\}_{n=1}^N$ sampled i.i.d. from $\mathcal{D}$, our approach is to use duality to approximate (P-CON) by the following empirical, unconstrained saddle point problem

$$\hat{D}^\star = \max_{\nu \geq 0} \min_{\boldsymbol{\theta} \in \Theta} \frac{1}{N} \sum_{n=1}^n \left[ \max_{\boldsymbol{\delta} \in \Delta} \ell\big(f_{\boldsymbol{\theta}}(\mathbf{x}_n + \boldsymbol{\delta}), y_n\big) + \nu \left[ \ell(f_{\boldsymbol{\theta}}(\mathbf{x}_n), y_n) - \rho \right] \right]. \tag{$\widehat{\text{DI}}$}$$

Conditions under which solutions of ($\widehat{\text{DI}}$) are (probably approximately) near-optimal *and* near-feasible for (P-CON) were obtained in [60]. As one would expect, these guarantees only hold when the objective and constraint of (P-CON) are learnable individually. As we discussed in Section 2, this is known to hold in a variety of scenarios (e.g., when the Rademacher complexity or VC-dimension is bounded), although obtaining more general results remains an area of active research [67–71]. In what follows, we formalize these generalization results in our setting, starting with the learning theoretic assumptions we require on the objective and constraint.

**Learning theoretic assumptions for** (P-CON). We first assume that the parameterization space $\Theta$ is sufficiently expressive (Assumption 4.1 and that there exists parameters $\boldsymbol{\theta} \in \Theta$ that strictly satisfy the nominal performance constraint (Assumption 4.2). We also assume that uniform convergence holds for the objective and constraint (Assumption 4.3).

**Assumption 4.1.** *The parametrization $f_{\boldsymbol{\theta}}$ is rich enough so that for each $\boldsymbol{\theta}_1, \boldsymbol{\theta}_2 \in \Theta$ and $\beta \in [0, 1]$, there exists $\boldsymbol{\theta} \in \Theta$ such that $\sup_{\mathbf{x} \in \mathcal{X}} |\beta f_{\boldsymbol{\theta}_1}(\boldsymbol{x}) + (1 - \beta) f_{\boldsymbol{\theta}_2}(\boldsymbol{x}) - f_{\boldsymbol{\theta}}(\boldsymbol{x})| \leq \alpha$.*

**Assumption 4.2.** *There exists $\boldsymbol{\theta}' \in \Theta$ such that $\mathbb{E}_{\mathcal{D}} \left[ \ell\big(f_{\boldsymbol{\theta}'}(\mathbf{x}), y\big) \right] < \rho - M\alpha$.*

**Assumption 4.3.** *There exists $\zeta_R(N), \zeta_N(N) \geq 0$ monotonically decreasing in $N$ such that $\forall \boldsymbol{\theta} \in \Theta$:*

$$\left| \mathbb{E}_{(\boldsymbol{x}, y) \sim \mathcal{D}} \left[ \max_{\boldsymbol{\delta} \in \Delta} \ell\big(f_{\boldsymbol{\theta}}(\mathbf{x} + \boldsymbol{\delta}), y\big) \right] - \frac{1}{N} \sum_{n=1}^N \max_{\boldsymbol{\delta} \in \Delta} \ell\big(f_{\boldsymbol{\theta}}(\boldsymbol{x}_n + \boldsymbol{\delta}), y_n\big) \right| \leq \zeta_R(N) \text{ w.p. } 1 - \delta \tag{8a}$$

$$\left| \mathbb{E}_{(\boldsymbol{x}, y) \sim \mathcal{D}} \left[ \ell\big(f_{\boldsymbol{\theta}}(\mathbf{x}), y\big) \right] - \frac{1}{N} \sum_{n=1}^N \ell(f_{\boldsymbol{\theta}}(\mathbf{x}_n), y_n) \right| \leq \zeta_N(N) \text{ w.p. } 1 - \delta \tag{8b}$$

One natural question to ask is whether the bounds in (8a) and (8b) hold in practice. We note that in the non-adversarial bound (8b) has been shown to hold for a wide variety of hypothesis classes, including DNNs [78, 58]. And although these classical results do not imply the robust uniform

---

**Algorithm 1** Semi-Infinite Dual Adversarial Learning (DALE)

---

Initialize $\boldsymbol{\theta} \leftarrow \boldsymbol{\theta}_0$ and $\nu \leftarrow 0$

1: **repeat**

2:     **for** Batch $\{(\mathbf{x}_i, y_i)\}_{i=1}^m$ **do**

3:         $\boldsymbol{\delta}_i \leftarrow \mathbf{0}$, for $i = 1, \ldots, m$

4:         **for** $L$ steps **do**

5:             $U_i \leftarrow \log\left[\ell_{\text{pert}}\big(f_{\boldsymbol{\theta}}(\mathbf{x}_i + \boldsymbol{\delta}_i), y_i\big)\right]$, for $i \in [m] := \{1, \ldots, m\}$

6:             $\boldsymbol{\delta}_i \leftarrow \prod_\Delta \left[\boldsymbol{\delta}_i + \eta \operatorname{sign}\left[\nabla_{\boldsymbol{\delta}_i} U_i + \sqrt{2\eta T}\boldsymbol{\xi}_i\right]\right]$, where $\boldsymbol{\xi}_i \sim \text{Laplace}(0, I), \ \ \forall i \in [m]$

7:         **end for**

8:         $\boldsymbol{\theta} \leftarrow \boldsymbol{\theta} - \dfrac{\eta_p}{m} \sum_{i=1}^m \nabla_{\boldsymbol{\theta}} \left[\ell_{\text{ro}}\big(f_{\boldsymbol{\theta}}(\mathbf{x}_i + \boldsymbol{\delta}_i), y_i\big) + \nu\ell_{\text{nom}}\big(f_{\boldsymbol{\theta}}(\mathbf{x}_i), y_i\big)\right]$

9:     **end for**

10:     $\nu \leftarrow \left[\nu + \eta_d \left(\dfrac{1}{N} \sum_{n=1}^N \ell\big(f_{\boldsymbol{\theta}}(\mathbf{x}_n), y_n\big) - \rho\right)\right]_+$

11: **until** convergence

---

convergence property (8a), there is a growing body of evidence which suggests that this property does in fact hold for the function class of DNNs [68, 79, 80].

**Near-optimality and near-feasibility of** $\widehat{(\text{DI})}$**.** By combining these assumptions with the techniques used in [60], we can explicitly bound the empirical duality gap (with high probability) and characterize the feasibility of the empircal dual optimal solution for (P-CON).

**Proposition 4.4** (The empirical dual of (P-CON))**.** *Let $\ell(\cdot, y)$ be a convex function for all $y \in \mathcal{Y}$. Under Assumptions 4.1–4.3, it holds with probability $1 - 5\delta$ that*

1. *$\left|P^\star - \hat{D}^\star\right| \leq M\alpha + (1 + \overline{\nu}) \max(\zeta_R(N), \zeta_N(N))$; and*

2. *There exists $\boldsymbol{\theta}^\dagger \in \operatorname{argmin}_{\boldsymbol{\theta} \in \Theta} \hat{L}(\boldsymbol{\theta}, \hat{\nu}^\star)$ such that $\mathbb{E}_{(\mathbf{x}, y) \sim \mathcal{D}} \left[\ell\big(f_{\boldsymbol{\theta}^\dagger}(\mathbf{x}), y\big)\right] \leq \rho + \zeta_N(N)$.*

*Here, $\hat{\nu}^\star$ denotes a solution of $\widehat{(\text{DI})}$, $\nu^\star$ denotes an optimal dual variable of (P-CON) solved over $\overline{\mathcal{H}} = \operatorname{conv}(\mathcal{H})$ instead of $\mathcal{H}$, and $\overline{\nu} = \max(\hat{\nu}^\star, \nu^\star)$. Additionally, for any interpolating classifier $\boldsymbol{\theta}'$, i.e. such that $\mathbb{E}_{(\mathbf{x}, y) \sim \mathcal{D}} \left[\ell\big(f_{\boldsymbol{\theta}'}(\mathbf{x}), y\big)\right] = 0$, it holds that*

$$\nu^\star \leq \rho^{-1} \mathbb{E}_{(\mathbf{x}, y) \sim \mathcal{D}} \left[\max_{\boldsymbol{\delta} \in \Delta} \ell\big(f_{\boldsymbol{\theta}'}(\mathbf{x} + \boldsymbol{\delta}), y\big)\right]. \tag{9}$$

The proof is provided in Appendix D. At a high level, Proposition 4.4 tells us that it is possible to learn robust models with high clean accuracy using the empirical dual problem in $\widehat{(\text{DI})}$ at little cost to the sample complexity. This means that seeking a robust classifier with a given nominal performance is (probably approximately) equivalent to seeking a classifier that minimizes a combination of the nominal and adversarial empirical loss. Notably, the majority of past approaches for solving (P-CON) cannot be endowed with similar guarantees in the spirit of Proposition 4.4. Indeed, while the objective resembles a penalty-based formulation, notice that $\nu$ is an *optimization variable* rather than a fixed hyperparameter. Concretely, the magnitude of this dual variable $\nu$ quantifies how hard it is to learn an adversarially robust model while maintaining strong nominal performance. Though seemingly innocuous, this caveat is the difference between guaranteeing generalization only on the aggregated loss and guaranteeing generalization jointly for the objective value and constraint feasibility.

## 5 Dual robust learning algorithm

Under the mild assumption that $(\mathbf{x}, y) \mapsto \ell(f_{\boldsymbol{\theta}}(\mathbf{x}), y) \in L^2$, Propositions 3.1, 3.2, and 4.4 allow us to transform (P-CON) into the following **D**ual **A**dversarial **LE**arning problem

$$\hat{D}^\star \triangleq \max_{\nu \geq 0} \min_{\boldsymbol{\theta} \in \Theta} \frac{1}{N} \sum_{n=1}^n \left[\mathbb{E}_{\boldsymbol{\delta}_n} \left[\ell(f_{\boldsymbol{\theta}}(\mathbf{x}_n + \boldsymbol{\delta}_n), y_n)\right] + \nu \left[\ell\big(f_{\boldsymbol{\theta}}(\mathbf{x}_n), y_n\big) - \rho\right]\right] \quad \text{(P-DALE)}$$

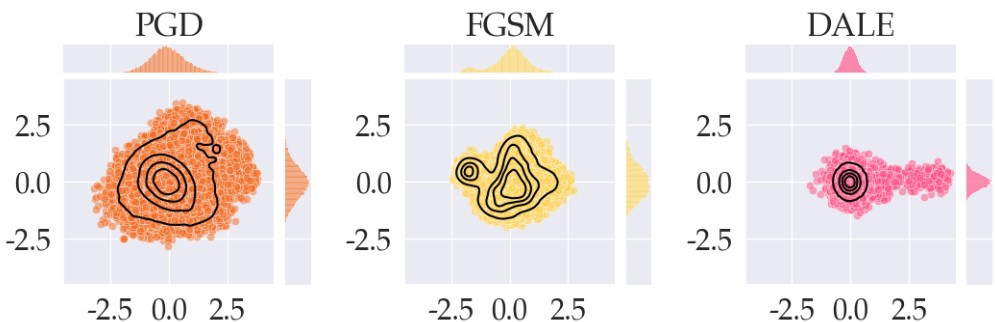

Figure 1: **Visualizing the distribution of adversarial perturbations.** In this figure, we visualize the distribution of adversarial perturbations by projecting the perturbations generated by PGD, FGSM, and DALE onto their first two principal components. The first and second principal components are shown on the $x$- and $y$-axes respectively. Notice that DALE varies much less along the second principal component vis-a-vis PGD and FGSM; this indicates that DALE tends to focus more on directions in which the data varies most, indicating that it finds stronger adversarial perturbations.

where $\boldsymbol{\delta}_n \sim \lambda_n^\star := \gamma_n^{-1}\big[\ell(f_{\boldsymbol\theta}(\mathbf{x}_n + \boldsymbol{\delta}_n), y_n) - \mu_n\big]_+$ for each $n \in \{1, \ldots, N\}$ and $\gamma_n > 0$ and $\mu_n$ are the constants specified in Proposition 3.2. Note that this formulation is considerably more amenable than (P-CON). Indeed, it is (i) empirical and therefore does not involve unknown statistical quantities such as $\mathcal{D}$; (ii) unconstrained and therefore more amendable to gradient-based optimization techniques; and (iii) its objective does not involve a challenging maximization problem in view of the closed-form characterization of $\lambda^\star$ in Proposition 3.2. In fact, for models that are linear in $\boldsymbol\theta$ but nonlinear in the input (e.g., kernel models or logistic regression), this implies that we can transform a non-convex, composite optimization problem (P-CON) into a convex problem (P-DALE).

Nevertheless, for many modern ML models such as CNNs, (P-DALE) remains a non-convex program in $\boldsymbol\theta$. And while there is overwhelming theoretical and empirical evidence that stochastic gradient-based algorithms yield good local minimizers for such overparametrized problems [35–39], the fact remains that solving (P-DALE) requires us to evaluate an expectation with respect to $\lambda^\star$, which is challenging due to the fact that $\mu_n$ and $\gamma_n$ are not known a priori. In the remainder of this section, we propose a practical algorithm to solve (P-DALE) based on the approximation discussed in Section 3.

**Sampling from the optimal distribution $\lambda_n^\star$.** Although Proposition 3.2 provides a characterization of the optimal distribution $\lambda_n^\star$, obtaining samples from $\lambda_n^\star$ can still be challenging in practice, especially when the dimension of $\boldsymbol{\delta}_n$ is large (e.g., for image-classification tasks). Moreover, in practice the value of $\gamma$ for which (5) is a solution of (4) is not known *a priori* and can be arbitrarily close to zero, making $\lambda_n^\star$ discontinuous and with a potentially vanishing support. Fortunately, these issues can be addressed by using Hamiltonian Monte Carlo (HMC) methods, which leverage the geometry of the distribution to overcome the curse of dimensionality.

In particular, we propose to use a projected Langevin Monte Carlo (LMC) sampler [81]. To derive this sampler, we first make a simplifying assumption: Rather than seeking the optimal constants $\gamma$ and $\mu$, we consider the over-smoothed approximation of $\lambda_n^\star$ derived in (7), wherein the probability mass allocated to a particular perturbation $\boldsymbol{\delta} \in \Delta$ is proportional to the loss $\ell(f_{\boldsymbol\theta}(\mathbf{x} + \boldsymbol{\delta}), y)$. We note that while this choice of $\lambda_n^\star$ may not be optimal, the sampling scheme that we derive under this assumption yields strong numerical performance. Furthermore, even if we knew the true values of $\gamma_n$ and $\mu_n$, the resulting distribution for $\mu_n \neq 0$ would be discontinuous and sampling from such distributions in high-dimensional settings is challenging in and of itself (see, e.g., [82]).

Given this approximate characterization of the optimal distribution, the following Langevin iteration can be derived directly from the commonly-used leapfrog simpletic integrator for the Hamiltonian dynamics induced by the distribution $\lambda_n$ defined in (7) (see Appendix E for details). This, in turn, yields the following update rule:

$$\boldsymbol{\delta} \leftarrow \Pi_\Delta \Big[ \boldsymbol{\delta} + \eta \operatorname{sign} \Big[ \nabla_{\boldsymbol{\delta}} \log \Big[ \ell_{\text{pert}}(f_{\boldsymbol\theta}(\mathbf{x} + \boldsymbol{\delta}), y) \Big] + \sqrt{2\eta T} \boldsymbol{\xi} \Big] \Big] \tag{10}$$

Table 1: **Adversarial robustness on MNIST and CIFAR-10.** Test accuracies of DALE (Algorithm 1) and state-of-the-art baselines on MNIST and CIFAR-10. On both datasets, DALE surpasses the baselines against both adversaries, while simultaneously maintaining high nominal performance.

| Algorithm | $\rho$ | MNIST | | | CIFAR-10 | | |
|---|---|---|---|---|---|---|---|
| | | Clean | FGSM | PGD$^{10}$ | Clean | FGSM | PGD$^{20}$ |
| ERM | - | 99.3 | 14.3 | 1.46 | 94.0 | 0.01 | 0.01 |
| FGSM | - | 98.3 | 98.1 | 13.0 | 72.6 | 49.7 | 40.7 |
| PGD | - | 98.1 | 95.5 | 93.1 | 83.8 | 53.7 | 48.1 |
| CLP | - | 98.0 | 95.4 | 92.2 | 79.8 | 53.9 | 48.4 |
| ALP | - | 98.1 | 95.5 | 92.5 | 75.9 | 55.0 | 48.8 |
| TRADES | - | 98.9 | 96.5 | 94.0 | 80.7 | 55.2 | 49.6 |
| MART | - | 98.9 | 96.1 | 93.5 | 78.9 | 55.6 | 49.8 |
| DALE | 0.5 | 99.3 | 96.6 | 94.0 | 86.0 | 54.4 | 48.4 |
| DALE | 0.8 | 99.0 | 96.9 | 94.3 | 85.0 | 55.4 | 50.1 |
| DALE | 1.0 | 99.1 | 97.7 | 94.5 | 82.1 | 55.2 | 51.7 |

where $\boldsymbol{\xi} \sim \text{Laplace}(\mathbf{0}, \boldsymbol{I})$. In this notation, $T > 0$ and $\eta > 0$ are constants which can be chosen as hyperparameters, and $\ell_{\text{pert}}$ is a loss functions for the perturbation. The resulting algorithm is summarized in Algorithm 1. Notice that Algorithm 1 accounts for scenarios in which the losses associated with the adversarial performance ($\ell_{\text{ro}}$), the perturbation ($\ell_{\text{pert}}$), and the nominal performance ($\ell_{\text{nom}}$) are different. It can therefore learn from perturbations that are adversarial for a different loss than the one used for training the model $\boldsymbol{\theta}$. This generality allows it to tackle different applications, e.g., by replacing the adversarial error objective in (P-CON) by a measure of model invariance (e.g., ACE in [53]). This feature can also be used to show that existing adversarial training procedures can be seen as approximations of Algorithm 1 (see Appendix A). We refer the reader to Appendix H for further discussion of the convergence properties of Algorithm 1.

## 6 Experiments

In this section, we include an empirical evaluation of the DALE algorithm. In particular, we consider two standard datasets: MNIST and CIFAR-10. For MNIST, we train four-layer CNNs and set $\Delta = \{\boldsymbol{\delta} : \|\boldsymbol{\delta}\|_\infty \leq 0.3\}$; for CIFAR-10, we train ResNet-18 models and set $\Delta = \{\boldsymbol{\delta} : \|\boldsymbol{\delta}\|_\infty \leq 8/255\}$. All hyperparameters and performance metrics are chosen with respect to the robust accuracy of a PGD adversary evaluated on a small hold-out validation set. Further details concerning hyperparameters and architectures are provided in Appendix F. We also provide additional experiments in Appendix G.

**Evaluating the adversarial robustness of DALE.** We begin our empirical evaluation by comparing the adversarial robustness of DALE with numerous state-of-the-art baselines in Table 1. To evaluate the robust performance of these classifiers, we use a 1-step and an $L$-step PGD adversary to evaluate robust performance; we denote these adversaries by FGSM and PGD$^L$ respectively. Notice that on CIFAR-10, DALE with $\rho = 0.8$ is the only method to achieve higher than 85% clean accuracy and 50% adversarial accuracy against PGD$^{20}$. Furthermore, when DALE is run with $\rho = 1.1$, we see that it achieves nearly 52% adversarial accuracy, which is a significant improvement over all baselines. In Appendix G, we provide a more complete characterization of the role of $\rho$ in controlling the trade-off between robustness and accuracy.

**Visualizing the distribution of adversarial perturbations.** To visualize the distribution over perturbations generated by DALE, we use principal component analysis (PCA) to embed perturbations into a two-dimensional space. In particular, we performed PCA on the MNIST training set to extract the first two principal components of the images; we then projected the perturbations $\boldsymbol{\delta} \in \Delta$ generated by PGD, FGSM, and DALE in the last iteration of training onto these principal components. A plot of these projections is shown in Figure 1, in which the first and second principal components are shown on the $x$- and $y$-axes respectively. Notice that the perturbations generated by FGSM are spread out somewhat unevenly in this space. In contrast, the perturbations found by PGD and DALE are spread out more evenly. Furthermore, the perturbations generated by PGD and FGSM vary more

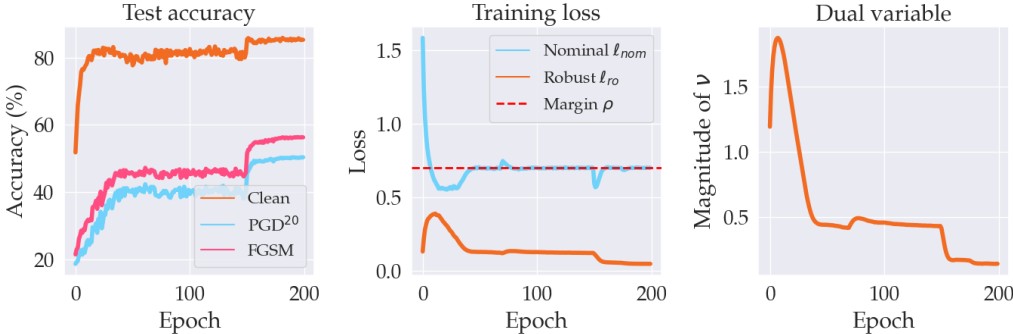

Figure 2: **Tracking the dual variables.** Left: the clean and robust test accuracies of a ResNet-18 classifier trained on CIFAR-10 using DALE. Middle: the training losses for DALE. Right: The magnitude of the dual variable during training.

along the second principal component ($y$-axis) than the first ($x$-axis) relative to DALE. Since the first component describes the direction of largest variance of the data, this indicates that DALE tends to find perturbations that place more mass on the direction in which the data varies most.

**Tracking the dual variables.** In Figure 2 we study the performance of DALE over the course of training. In the leftmost panel, we plot the test accuracy on clean samples and on adversarially perturbed samples. Notably, this classifier exceeds 50% robust accuracy against PGD$^{20}$ as well as 85% clean accuracy; these figures are higher than either of the corresponding metrics for any of the baselines in Table 1, indicating that our method is more effectively able to mitigate the trade-off between robustness and accuracy. In the middle panel of Figure 2, we track the nominal and robust training losses, and in the rightmost panel, we show the magnitude of the dual variable $\nu$. Observe that at the onset of training, the constraint in (P-CON) is not satisfied, as the blue curve is above the red dashed-line. In response, the dual variable places more weight on the nominal loss term in (P-DALE). After several epochs, this reweighting forces constraint satisfaction, after which the dual variable begins to decrease, which in turn decreases the weight on the nominal objective and allows the optimizer to focus on minimizing the robust loss.

**Regularization vs. primal-dual.** Our final ablation study is to consider the impact of performing the dual-update step in line 10 of Algorithm 1. In particular, in Table 2, we record the performance of DALE when Algorithm 1 is run without the dual update step. This corresponds to running DALE with a fixed weight $\nu$. Notice that although our method reaches the same level of robust performance as MART and TRADES, it does not match the performance of the DALE classifiers in Table 1. This indicates that the strong robust performance of our algorithm relies on adaptively updating the dual variable over the course of training.

Table 2: **Regularized DALE.** Test accuracies attained by running DALE without the dual-update step in line 10 of Algorithm 1.

| $\nu$ | Clean | FGSM | PGD$^{20}$ |
|---|---|---|---|
| 0.1 | 86.4 | 55.3 | 49.5 |
| 0.2 | 86.8 | 54.2 | 49.3 |
| 0.3 | 86.3 | 54.8 | 48.2 |
| 0.4 | 86.2 | 54.6 | 47.3 |
| 0.5 | 86.5 | 54.3 | 46.8 |
| 0.6 | 85.7 | 53.3 | 46.4 |
| 0.7 | 85.8 | 53.3 | 46.0 |
| 0.8 | 84.9 | 53.1 | 45.9 |
| 0.9 | 85.0 | 53.4 | 45.7 |
| 1.0 | 84.5 | 52.7 | 45.8 |

## 7   Conclusion

In this paper, we studied robust learning from a constrained learning perspective. We proved an equivalence between the standard adversarial training paradigm and a stochastic optimization problem over a specific, non-atomic distribution. This insight provided a new perspective on robust learning and engendered a Langevin MCMC approach for adversarial robustness. We experimentally validated that this algorithm outperforms the state-of-the-art on standard benchmarks. Notably, our method simultaneously achieved greater than 50% adversarial accuracy and greater than 85% clean accuracy on CIFAR-10, which represents a significant improvement over the previous state-of-the-art.

## 8 Acknowledgements and disclosure of funding

The authors would like to thank Juan Cervino and Samuel Sokota for their helpful feedback.

The research of Alexander Robey and Hamed Hassani is supported by NSF Grants 1837253, 1943064, 1934876, AFOSR Grant FA9550-20-1-0111, and DCIST-CRA. Alexander Robey, George J. Pappas, Luiz F. O. Chamon, and Alejandro Ribeiro are all supported by the NSF-Simons Program on the Mathematics of Deep Learning. Alexander Robey and George J. Pappas are supported by ARL CRA DCIST W911NF-17-2-0181 and NSF 2038873.

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
