# A    Connections to other problems

In the Introduction, we claimed that several well-known formulations for robust learning can be recovered as approximations of (PII) or Algorithm 1. In what follows, we explore these connections in three directions: (i) fixed, sub-optimal choices of the perturbation distribution $\lambda$ in (4); (ii) limiting cases of the projected LMC dynamics used in Algorithm 1; (iii) modifications of the empirical dual problem $\widehat{(\text{DI})}$.

## A.1    Sub-optimal perturbation distributions

For a fixed perturbation distribution $\lambda$ in (4), (PII) can be thought of as a random data augmentation procedure [83]. For instance, for $\lambda = \mathcal{N}(\mathbf{0}, \mathbf{\Sigma})$, (PII) becomes

$$\underset{\boldsymbol{\theta} \in \Theta}{\text{minimize}} \; \mathbb{E}_{(\mathbf{x},y) \sim \mathcal{D}} \left[ \mathbb{E}_{\boldsymbol{\delta} \sim \mathcal{N}(\mathbf{0}, \mathbf{\Sigma})} \left[ \ell\big(f_{\boldsymbol{\theta}}(\mathbf{x} + \boldsymbol{\delta}), y\big) \right] \right].$$

Indeed, the authors of [84, 85] suggest that Gaussian data augmentation can significantly improve generalization, particularly the augmentations are applied using patches [84]. To quote from [85]:

> Data augmentation with Gaussian ... noise serves as a simple yet very strong baseline that is sufficient to surpass almost all previously proposed defenses against common corruptions.

More complex choices of distributions lead to other robust learning methods involving random removal of patches from the image (e.g., `Cutout` [86, 87]), random replacement of patches (e.g., `CutMix` [88, 89]), or arbitrary generative models, i.e., $\boldsymbol{\delta} \sim G \,\#\, \mathcal{N}(\mathbf{0}, \sigma^2 \boldsymbol{I})$ for some measurable $G : \mathbb{R}^z \to \mathbb{R}^d$ [85]. For generative models with latent dimension $z \ll d$, the latter approach can be thought of as parameterizing the perturbation distribution $\lambda$ on a lower-dimensional manifold in the data space, which has been shown to be a strong defense in prior work [90–92].

While data augmentation with random noise has been shown to be an effective method for improving robustness in practice, the results in this paper show that even larger gains are possible by optimizing over the perturbation-generating distribution. In particular, Proposition 3.2 establishes that the optimal perturbation distribution is not Gaussian and most importantly, not isotropic. Indeed, Figure 1 suggests that the pertubation distribution arising from Algorithm 1 does resemble an anisotropic Gaussian, but only on the basis induced by the principal components of the data.

It is worth noting that, as we mentioned in Section 2, the results of this paper do not rely on the linearity of the perturbations. Hence, more complex pertubations can be considered by using an arbitrary, parametrized data transformation $G : \mathcal{X} \times \Delta \to \mathcal{X}$ as in

$$\underset{\boldsymbol{\theta} \in \Theta}{\text{minimize}} \; \mathbb{E}_{(\mathbf{x},y) \sim \mathcal{D}} \left[ \mathbb{E}_{\boldsymbol{\delta} \sim \lambda} \left[ \ell\big(f_{\boldsymbol{\theta}}\big(G(\mathbf{x}, \boldsymbol{\delta})\big), y\big) \right] \right]. \tag{PIV}$$

Due to space constraints, we considered only pertubations of the form $G(\mathbf{x}, \boldsymbol{\delta}) = \mathbf{x} + \boldsymbol{\delta}$ as in (P-RO). Yet, by once again fixing the pertubation distribution $\lambda$, we can obtain a myriad of data augmentation techniques, including the group-theoretic data-augmentation scheme discussed in [93], where $G$ denotes the group action, and the model-based robust training methods discussed in [62–65]. Indeed, exploring the efficacy of DALE toward improving robustness beyond norm-bounded perturbations is an exciting direction for future work.

## A.2    Sampling vs. optimizing pertubations

Aside from fixing the pertubation distribution $\lambda$, another common approach to adversarial learning is to use a gradient-based local optimization method in order to tackle the maximization in $\ell_{\text{adv}}$ (see e.g., [26, 27]). The perturbations found by these gradient-based methods can then be used to train a robust model. While empirically effective, this approach is not without issues. In particular, gradient-based algorithms are not guaranteed to obtain optimal (or even near-optimal) perturbations, since $\ell(f_{\boldsymbol{\theta}}(\cdot), y)$ is typically not a convex (or concave) function. What is more, maximizing over $\boldsymbol{\delta}$ in the definition of $\ell_{\text{adv}}$ is a severely underparametrized problem as opposed to the minimization over $\boldsymbol{\theta}$ in (P-RO). It therefore does not enjoy the same benign optimization landscape [35–39]. Additionally, note that there is no guarantee that this alternating optimization technique converges.

Nevertheless, these algorithms can be seen as limiting cases of Algorithm 1 for specific choices of the losses $\ell_{\text{pert}}$, $\ell_{\text{ro}}$, and $\ell_{\text{nom}}$, the LMC kinetic energy (step 6), and the temperature ($T$ in step 5). To illustrate this idea, suppose that both $\ell_{\text{pert}}$ and $\ell_{\text{ro}}$ are taken to be the cross-entropy loss, i.e.,

$$\ell\big(f_{\boldsymbol{\theta}}(\mathbf{x}), y\big) = -\log\big([f_{\boldsymbol{\theta}}(\mathbf{x})]_y\big). \tag{11}$$

In this case, when we take $T \to 0$, Algorithm 1 approaches the gradient-based attacks FGSM (for $L = 1$) [26] and PGD (for $L > 1$) [27]. However, as we observed in Figure 1, these methods can produce quite different perturbations compared to the perturbations produced by DALE.

Another interesting perspective on gradient-based methods is to consider a different sampling scheme. Indeed, while we adopted the commonly used Laplacian LMC sampler in Algorithm 1, an alternative often used to sample from lighter tailed distributions is Gaussian LMC [94, 81, 82]. In the latter, rather than defining the kinetic energy of the Hamiltonian as $K(\boldsymbol{p}) \propto \|\boldsymbol{p}\|_1$, this prior is taken to be $K(\boldsymbol{p}) \propto \|\boldsymbol{p}\|_2^2$. This is equivalent to replacing steps 5 and 6 of Algorithm 1 by

$$U \leftarrow \log\Big[\ell_{\text{pert}}(f_{\boldsymbol{\theta}}(\mathbf{x} + \boldsymbol{\delta}), y)\Big] \tag{12}$$

$$\boldsymbol{\delta} \leftarrow \Pi_\Delta\Big[\boldsymbol{\delta} + \nabla_{\boldsymbol{\delta}} U + \sqrt{2\eta T}\boldsymbol{\xi}\Big] \tag{13}$$

where $\boldsymbol{\xi} \sim \mathcal{N}(\mathbf{0}, \boldsymbol{I})$. An interesting direction for future work is to compare the performance of HMC-based samplers under different priors. For more details regarding the derivation of our LMC sampler, see Appendix E.

### A.3 Penalty-based methods

The third approximation of Algorithm 1, or more precisely, $\widehat{(\text{DI})}$, is the use of a fixed $\nu > 0$, e.g., [53, 54, 95]. Indeed, notice that in the definition of $\hat{L}$, $\nu$ is an optimization variable that is dynamically adjusted in Algorithm 1 through the dual ascent update in step 10. Notice that step 10 is simply a (sub)gradient ascent update given that the constraint violation is a subgradient of the dual function $\hat{d}(\nu) = \min_{\boldsymbol{\theta} \in \Theta} \hat{L}(\boldsymbol{\theta}, \nu)$ for the empirical Lagrangian (see Lemma D.2).

While effective, there are clear advantages in letting $\nu$ be an optimization variable. In practice, not only does it lead to improved performance (see Section 6), but it has the advantage of precluding the need to manually adjust another hyperparameter, which can be challenging and often requires domain-specific knowledge. Indeed, the value of $\nu$ depends on the underlying learning task (model, losses, dataset), making it difficult to transfer across applications and highly dependent on domain knowledge. What is more, if not done carefully, it can hinder generalization guarantees for the solution.

This issue is, in fact, at the core of the theoretical advantage of $\widehat{(\text{DI})}$. Indeed, note that classical learning theory [57, 58] provides generalization bounds only for the aggregated objective and not each individual penalty term, i.e., for the value of the Lagrangian rather than the adversarial and nominal losses in (P-CON). In contrast, Proposition 4.4 provides generalization guarantees both in terms of near-optimality and near-feasibility by leveraging the constrained learning theory developed in [60, 77].

# B  Proof of Proposition 3.1

Start by writing the primal problem (PI) in Lagragian form [96, Ch. 4]. Explicitly,

$$P_R^\star = \min_{\boldsymbol{\theta}\in\Theta, t\in L^p} \max_{\overline{\lambda}\in L_+^q} L_{\text{PI}}(\boldsymbol{\theta}, t, \overline{\lambda}), \tag{PV}$$

where $L_+^q$ denotes the subspace of almost everywhere non-negative functions of $L^q$ for $(1/p) + (1/q) = 1$. Here the Lagrangian $L_{\text{PI}}(\boldsymbol{\theta}, t, \overline{\lambda})$ is defined as

$$L_{\text{PI}}(\boldsymbol{\theta}, t, \overline{\lambda}) = \mathbb{E}_{(\mathbf{x},y)\sim\mathcal{D}}\big[t(\mathbf{x},y)\big] + \int \overline{\lambda}(\mathbf{x},\boldsymbol{\delta},y)\big[\ell\big(f_{\boldsymbol{\theta}}(\mathbf{x}+\boldsymbol{\delta}),y\big) - t(\mathbf{x},y)\big]\,d\mathbf{x}d\boldsymbol{\delta}dy$$

$$= \int t(\mathbf{x},y)\Big[\mathfrak{p}(\mathbf{x},y) - \int \overline{\lambda}(\mathbf{x},\boldsymbol{\delta},y)d\boldsymbol{\delta}\Big]\,d\mathbf{x}dy + \int \overline{\lambda}(\mathbf{x},\boldsymbol{\delta},y)\ell\big(f_{\boldsymbol{\theta}}(\mathbf{x}+\boldsymbol{\delta}),y\big)d\mathbf{x}d\boldsymbol{\delta}dy, \tag{14}$$

where we used the density $\mathfrak{p}$ of the data distribution $\mathcal{D}$. Then, notice that (PV) can be written iteratively as

$$P_R^\star = \min_{\boldsymbol{\theta}\in\Theta} p(\boldsymbol{\theta}) \quad \text{where} \quad p(\boldsymbol{\theta}) = \min_{t\in L^p} \max_{\overline{\lambda}\in L_+^q} L_{\text{PI}}(\boldsymbol{\theta}, t, \overline{\lambda}). \tag{PVI}$$

Observe that $\boldsymbol{\theta}$ is constant in the definition of $p(\boldsymbol{\theta})$. Since (14) is a linear function of $t$, $p(\boldsymbol{\theta})$ is the optimal value of a linear program parametrized by $\boldsymbol{\theta}$. Hence, strong duality holds [96, Ch. 4] and we obtain that

$$p(\boldsymbol{\theta}) = \max_{\overline{\lambda}\in L_+^q} d_{\text{PI}}(\overline{\lambda}) \quad \text{where} \quad d_{\text{PI}}(\overline{\lambda}) = \min_{t\in L^p} L_{\text{PI}}(\boldsymbol{\theta}, t, \overline{\lambda}), \tag{15}$$

for the dual function $d_{\text{PI}}$. Since $t$ is unconstrained and $L$ is linear in $t$, the dual function either vanishes for $\mathfrak{p}(\mathbf{x},y) = \int \overline{\lambda}(\mathbf{x},\boldsymbol{\delta},y)d\boldsymbol{\delta}$ or diverges to $-\infty$. From (14) and (15), we thus obtain that

$$p(\boldsymbol{\theta}) = \max_{\overline{\lambda}\in L_+^q} \quad \int \overline{\lambda}(\mathbf{x},\boldsymbol{\delta},y)\ell\big(f_{\boldsymbol{\theta}}(\mathbf{x}+\boldsymbol{\delta}),y\big)d\mathbf{x}d\boldsymbol{\delta}dy$$

$$\text{subject to} \quad \int \overline{\lambda}(\mathbf{x},\boldsymbol{\delta},y)d\boldsymbol{\delta} = \mathfrak{p}(\mathbf{x},y) \tag{16}$$

To conclude, notice that since $\overline{\lambda}$ is almost everywhere non-negative, it must be that $\overline{\lambda}(\mathbf{x},\boldsymbol{\delta},y) = 0$ for all $\boldsymbol{\delta}\in\Delta$ whenever $\mathfrak{p}(\mathbf{x},y) = 0$. The measure induced by $\overline{\lambda}$ is therefore absolutely continuous with respect $\mathcal{D}$. We can therefore rewrite (16) in terms of the Radon-Nykodim derivative,

$$\lambda(\boldsymbol{\delta} \mid \mathbf{x},y) = \overline{\lambda}(\mathbf{x},\boldsymbol{\delta},y)/\mathfrak{p}(\mathbf{x},y) \tag{17}$$

which yields (4) as desired.

## C Proof of Proposition 3.2

For the sake of completeness, before proving Proposition 3.2, in Section C.1 we provide a short discussion of the preliminary material needed to prove the proposition. The majority of this exposition is adapted from Rockafellar and Wets' *Variational Analysis* [97]. Following this, in Section C.2 we present a lemma which establishes the decomposability of $\mathcal{P}^q$ over $\Omega$, which is crucial in proving the proposition. Finally, in Section C.3 we provide the full proof of Proposition 3.2.

### C.1 Preliminaries

Throughout these preliminaries, we let the tuple $(T, \mathcal{A})$ denote a measurable space, where $T$ is a nonempty set and $\mathcal{A}$ is a $\sigma$-algebra of measurable sets belonging to $T$. Furthermore, we let $\overline{\mathbb{R}}$ denote the extended real-line, and we will use $\mu$ to refer to an arbitrarily defined measure over the measurable space $(T, \mathcal{A})$.[4] By $\mathcal{G}$ we denote an arbitrary space of measurable functions $g : T \to \mathbb{R}^n$. To this end, given an integrand $f : T \times \mathbb{R}^n \to \overline{\mathbb{R}}$, we will consider integral functionals of the form

$$I_f[g] = \int_T f(t, g(t))\mu(dt) \tag{18}$$

To begin our preliminaries, we first recall the definition of a normal integrand.

**Definition C.1** (Normal integrand). *A function $f : T \times \mathbb{R}^n \to \overline{\mathbb{R}}$ is called a **normal integrand** if its epigraphical mapping $S_f : T \to \mathbb{R}^n \times \mathbb{R}$ defined by*

$$S_f(t) \triangleq epi f(t, \cdot) = \left\{ (x, \alpha) \in \mathbb{R}^n \times \mathbb{R} \,\middle|\, f(t, x) \le \alpha \right\} \tag{19}$$

*is closed valued and measurable.*

**Definition C.2** (Carathéodory integrand). *A function $f : T \times \mathbb{R}^n \to \mathbb{R}$ is called a **Carathéodory integrand** if it is measurable in $t$ for each $x$ and continuous in $x$ for each $t$.*

Notably, Carathéodory integrands are a special case of normal integrands (see e.g., [97, Ex. 14.29]). Next, recall the definition of a *decomposable space*.

**Definition C.3** (Decomposable space). *A space $\mathcal{G}$ of measurable functions $g : T \to \mathbb{R}^n$ is **decomposable** in association with a measure $\mu$ on $\mathcal{A}$ if for every function $g_0 \in \mathcal{G}$, for every set $A \in \mathcal{A}$ with $\mu(A) < \infty$, and for every bounded, measurable function $g_1 : A \to \mathbb{R}^n$, the space $\mathcal{G}$ contains the function $g : T \to \mathbb{R}^n$ defined by*

$$g(t) = \begin{cases} g_0(t) & \text{for } t \in T \backslash A, \\ g_1(t) & \text{for } t \in A. \end{cases} \tag{20}$$

Note that the space $\mathcal{M}(T, A)$ of measurable functions $g : T \to \mathbb{R}^n$ is decomposable, as are the Lebesgue spaces $L^p(T, \mathcal{A}, \mu)$ for all $p \in [1, \infty]$ (see e.g., [97, Ch. 14]). As we will see, the decomposability of the Lebesgue spaces is integral to the proof of Proposition 3.2. However, before proceeding to the proof, we first restate a crucial result concerning the interchangability of minimization and integration, which relies on this notion of decomposability defined above.

**Theorem C.4** (Thm. 14.60 in [97]). *Let $\mathcal{G}$ be a space of measurable functions from $T$ to $R^n$ that is decomposable relative to a $\sigma$-finite measure $\mu$ defined on $\mathcal{A}$. Let $f : T \times \mathbb{R}^n$ be a normal integrand. Then the minimization of $I_f$ over $\mathcal{G}$ can be reduced to a pointwise minimization in the sense that, as long as $I_f \not\equiv 0$ on $\mathcal{G}$, one has*

$$\inf_{g \in \mathcal{G}} \int_T f(t, g(t))\mu(dt) = \int_T \left[ \inf_{x \in \mathbb{R}^n} f(t, x) \right] \mu(dt) \tag{21}$$

*Moreover, as long as this common value is not $-\infty$, one has for $\overline{g} \in \mathcal{G}$ that*

$$\overline{g} \in \operatorname*{argmin}_{g \in \mathcal{G}} I_f[g] \iff \overline{g}(t) \in \operatorname*{argmin}_{x \in \mathbb{R}^n} f(t, x) \quad \text{for } \mu\text{-almost every } t \in T. \tag{22}$$

The utility of this result is that under the assumptions that function class $\mathcal{G}$ is decomposable, the integrand $f$ is normal, and $I_f$ is finite over $\mathcal{G}$, it holds that the minimization and integration operations can be exchanged. Furthermore, note that this result is more general than we need; indeed, all of the integrands we work with in the next subsection are Carathéodory and hence normal.

---

[4]In some cases, we will also use $\mu$ to denote the Lebesgue measure on $\mathbb{R}^d$; this distinction will be made clear when we use this convention.

## C.2 A preliminary lemma

The first step toward proving Proposition 3.2 is to show that the space $\mathcal{P}_q$ is decomposable over the data space $\Omega$. We state this result in the following lemma, as it may be of expository interest as a warm-up before the proof of Proposition 3.2.

**Lemma C.5** (Decomposability of $\mathcal{P}^q$ over $\Omega$). *The space $\mathcal{P}^q$ of distributions of $\Delta$ defined in Proposition 3.2 is decomposable over $\Omega = \mathcal{X} \times \mathcal{Y}$ in the sense of definition C.3.*

*Proof.* Let $\mu$ denote the Lebesgue measure on $\mathbb{R}^d$. Recall that $\mathcal{P}^q$ is the subset of $L^p$ containing functions $\lambda$ with the following properties:

(P1) $\lambda(\cdot|\mathbf{x}, y)$ is almost everywhere non-negative on $\Omega$,

(P2) $\lambda(\cdot|\mathbf{x}, y)$ is absolutely continuous with respect to $\mathfrak{p}(\mathbf{x}, y)$,

(P3) $\int_\Delta \lambda(\boldsymbol{\delta}|\mathbf{x}, y)\mu(d\boldsymbol{\delta}) = 1$   for $\mathfrak{p}$-almost every $(\mathbf{x}, y) \in \Omega$.

To show that $\mathcal{P}^q$ is decomposable over $\Omega$, first let $\lambda, \lambda' \in \mathcal{P}^q$ and $A \subseteq \Omega$ with $\mu(A) < \infty$ be arbitrarily chosen. Define the functional

$$\overline{\lambda}(\boldsymbol{\delta}|\mathbf{x}, y) = \begin{cases} \lambda(\boldsymbol{\delta}|\mathbf{x}, y) & \text{for } (\mathbf{x}, y) \in \Omega \backslash A, \\ \lambda'(\boldsymbol{\delta}|\mathbf{x}, y) & \text{for } (\mathbf{x}, y) \in A. \end{cases} \tag{23}$$

Our goal is to show that $\overline{\lambda}$ is an element of $\mathcal{P}^q$. To begin, observe that by (P1), $\lambda$ and $\lambda'$ are almost everywhere non-negative, and therefore so is $\overline{\lambda}$. Further, by (P2), both $\lambda$ and $\lambda'$ are absolutely continuous with respect to $\mathfrak{p}$. Thus, observe that if $B \in \mathcal{B}$ such that $\mathfrak{p}(B) = 0$, then $\lambda(\delta|B) = \lambda'(\delta|B) = 0$, and thus it holds that $\overline{\lambda}(\delta|B) = 0$, proving that $\overline{\lambda} \ll \mathfrak{p}$. Finally, note that by (P3), both $\lambda$ and $\lambda'$ are normalized along $\Delta$. Thus, for any fixed $(\mathbf{x}, y) \in \Sigma$, it holds that $\int_\Delta \overline{\lambda}(\boldsymbol{\delta}|\mathbf{x}, y)\mu(d\delta) = 1$. Thus, it holds that $\overline{\lambda} \in \mathcal{P}^q$, as was to be shown. $\qquad\square$

## C.3 Proof of Proposition 3.2

Ultimately, there are three main steps to this proof. (1) First, we argue that Thm. C.4 applies to the maximization over $\lambda$, so that the expectation over the data distribution $\mathcal{D}$ and the maximization over $\lambda \in \mathcal{P}^q$ can be interchanged. (2) We argue that strong duality holds for the inner problem induced by pushing the maximization inside the expectation. (3) We find a closed-form solution for the dual problem, proving the claim of the proposition.

*Proof.* **Step 1.** To begin, we argue that the maximization over $\lambda$ and the expectation over the data distribution $\mathcal{D}$ can be interchanged. To do so, we define the function $F : \Omega \times \mathcal{P}^2 \to \mathbb{R}$

$$F\big((\mathbf{x}, y), \lambda\big) \triangleq \mathbb{E}_{\boldsymbol{\delta} \sim \lambda(\boldsymbol{\delta}|\mathbf{x}, y)} \big[\ell(f_{\boldsymbol{\theta}}(\mathbf{x} + \boldsymbol{\delta}), y)\big] \tag{24}$$

so that the optimization problem in (PII) can be written as

$$P_R^\star = \min_{\boldsymbol{\theta} \in \Theta} p(\boldsymbol{\theta}) \quad \text{where} \quad p(\boldsymbol{\theta}) \triangleq \max_{\lambda \in \mathcal{P}^2} \mathbb{E}_{(\mathbf{x}, y) \sim \mathcal{D}} \big[F\big((\mathbf{x}, y), \lambda\big)\big]. \tag{25}$$

Now observe that by construction $F$ is measurable in $(\mathbf{x}, y)$ for each $\lambda$, and continuous (in fact, linear) in $\lambda$ for each $(\mathbf{x}, y)$. Thus, $F$ is a Carathéodory integrand, and as $\mathcal{P}^2$ is decomposable over $\Omega$ by Lemma C.5, Thm. C.4 applies to $p(\boldsymbol{\theta})$. Therefore, the maximization in the definition of $p(\boldsymbol{\theta})$ can be pushed inside the expectation over $(\mathbf{x}, y) \sim \mathcal{D}$, yielding the following equality:

$$p(\boldsymbol{\theta}) = \max_{\lambda \in \mathcal{P}^2} \mathbb{E}_{(\mathbf{x}, y) \sim \mathcal{D}} \big[F\big((\mathbf{x}, y), \lambda\big)\big] = \mathbb{E}_{(\mathbf{x}, y) \sim \mathcal{D}} \Big[\max_{\lambda \in \mathcal{P}^2} F((\mathbf{x}, y), \lambda)\Big]. \tag{26}$$

**Step 2.** Next, we argue that the inner maximization problem in the expression on the RHS of (26) is strongly dual. To begin, notice that this inner maximization is now performed separately for each data point $(\mathbf{x}, y) \in \Omega$. We therefore proceed by considering the solution of the inner problem for an

arbitrary but fixed data point $(\overline{\mathbf{x}}, \overline{y})$. To this end, first let $\lambda^\star$ be the solution to the inner problems for this fixed pair $(\overline{\mathbf{x}}, \overline{y})$, i.e. $\lambda^\star$ achieves the optimal value in

$$u(\boldsymbol{\theta}) = u(\boldsymbol{\theta}, \overline{\mathbf{x}}, \overline{y}) = \max_{\lambda \in \mathcal{P}^2} F((\overline{\mathbf{x}}, \overline{y}), \lambda) \tag{27}$$

Now let $\mathfrak{m}(\Delta)$ denote the Lebesgue measure of $\Delta$, and consider that Hölder's inequality implies that $\|\lambda^\star\|_{L^1} \leq \mathfrak{m}(\Delta)^{1/2} \|\lambda^\star\|_{L^2}$. Further, as each feasible $\lambda \in \mathcal{P}^2$ is normalized over $\Delta$, it holds that

$$\frac{1}{\mathfrak{m}(\Delta)} \leq \|\lambda^\star\|_{L^2}^2. \tag{28}$$

Thus, since $\mathcal{P}^2 \subset L_+^2$, it holds that $\lambda^\star \in L_+^2$, and thus there exists a constant $c$ satisfying $1/\mathfrak{m}(\Delta) \leq c < \infty$ such that

$$\|\lambda^\star\|_{L^2}^2 = \int_\Delta F((\overline{\mathbf{x}}, \overline{y}), \delta)^2 d\delta \leq c. \tag{29}$$

Accordingly, we can rewrite (27) in an equivalent way as follows:

$$u(\boldsymbol{\theta}) = \max_{\lambda \in L_+^2} \quad \mathbb{E}_{\boldsymbol{\delta} \sim \lambda(\boldsymbol{\delta}|\mathbf{x}, y)} \left[ \ell(f_{\boldsymbol{\theta}}(\overline{\mathbf{x}} + \boldsymbol{\delta}), \overline{y}) \right]$$

$$\text{subject to} \quad \int_\Delta \lambda(\boldsymbol{\delta}|\overline{\mathbf{x}}, \overline{y}) d\delta = 1, \quad \int_\Delta \lambda(\boldsymbol{\delta}|\overline{\mathbf{x}}, \overline{y})^2 d\delta \leq c. \tag{PVII}$$

Notice that (PVII) is a convex quadratic program in the optimization variable $\lambda$. Furthermore, note that if $c = 1/\mathfrak{m}(\Delta)$ (i.e., equality is achieved in the expression (28) derived from Hölder's inequality), then the feasible set is a singleton which is equivalent in $L_2$ to $\lambda(\boldsymbol{\delta}|\overline{\mathbf{x}}, \overline{y}) = 1/\mathfrak{m}(\Delta)$. Alternatively, if $c > 1/\mathfrak{m}(\Delta)$, then $\lambda(\boldsymbol{\delta}|\overline{\mathbf{x}}, \overline{y})$ is a strictly feasible point, and thus Slater's condition holds. In either case, we find that (PVII) is strongly dual [96, Ch. 4] and so we can write

$$u(\boldsymbol{\theta}) = \min_{\gamma \geq 0, \, \mu \in \mathbb{R}} d_{\text{PVII}}(\boldsymbol{\theta}, \gamma, \mu), \tag{30}$$

for the dual function

$$d_{\text{PVII}}(\boldsymbol{\theta}, \gamma, \mu) = \max_{\lambda \in L_+^2} \int_\Delta \left[ \ell(f_{\boldsymbol{\theta}}(\overline{\mathbf{x}} + \boldsymbol{\delta}) \lambda(\boldsymbol{\delta}|\overline{\mathbf{x}}, \overline{y}) - \gamma \lambda(\boldsymbol{\delta}|\overline{\mathbf{x}}, \overline{y})^2 - \mu \lambda(\boldsymbol{\delta}|\overline{\mathbf{x}}, \overline{y}) \right] d\delta + \gamma c + \mu.$$

**Step 3.** Finally, we find a closed-form expression for the solution to the dual problem derived above. To do so, an entirely similar argument to the one given in Lemma C.5 shows that $L_+^2$ is decomposable. And indeed, as the integrand in the above primal function is clearly Carathéodory, we can again apply Theorem C.4 to $d_{\text{PVII}}(\boldsymbol{\theta}, \gamma, \mu)$:

$$d_{\text{PVII}}(\boldsymbol{\theta}, \gamma, \mu) = \int_\Delta \left\{ \max_{\lambda \in L_+^2} \ell(f_{\boldsymbol{\theta}}(\overline{\mathbf{x}} + \boldsymbol{\delta}) \lambda(\boldsymbol{\delta}|\overline{\mathbf{x}}, \overline{y}) - \gamma \lambda(\boldsymbol{\delta}|\overline{\mathbf{x}}, \overline{y})^2 - \mu \lambda(\boldsymbol{\delta}|\overline{\mathbf{x}}, \overline{y}) \right\} d\delta + \gamma c + \mu. \tag{31}$$

A straightforward calculation of the inner maximization problem shown above yields

$$\lambda^\star(\boldsymbol{\delta}|\overline{\mathbf{x}}, \overline{y}) = \left[ \frac{\ell(f_{\boldsymbol{\theta}}(\overline{\mathbf{x}} + \boldsymbol{\delta}), \overline{y}) - \mu}{2\gamma} \right]_+, \tag{32}$$

where $[z]_+ = \max(0, z)$ denotes the projection onto the non-negative orthant. From (PVII), $\mu$ is chosen so as to meet the normalization constraint, i.e., so that

$$\int_\Delta \left[ \frac{\ell(f_{\boldsymbol{\theta}}(\overline{\mathbf{x}} + \boldsymbol{\delta}), \overline{y}) - \mu}{2\gamma} \right]_+ d\delta = 1 \iff \int_\Delta [\ell(f_{\boldsymbol{\theta}}(\overline{\mathbf{x}} + \boldsymbol{\delta}), \overline{y}) - \mu]_+ d\delta = 2\gamma.$$

To conclude, notice that due to the strong duality of (PVII), for each value of $c < \infty$ there is a value of $\gamma > 0$ such that (32) is a solution of (PVII) [96, Ch. 4]. Also, since $(\overline{\mathbf{x}}, \overline{y})$ were chosen arbitrarily, (32) holds for all data point. Given that the space is decomposable, these solutions can be pieced together, yielding the desired result. □

# D  Proof of Proposition 4.4

We proceed here as in [60]. However, we deviate slightly from the proof of the parametrization gap [60, Prop. 2 in Appendix B.1] to account for the maximization in the robust loss. In particular, the proof of Proposition 3.6 is organized in the following way:

1. First, in Section D.1 we bound the deviation between the primal problem (P-CON) and dual problem (DI). The result is summarized in Lemma D.1.

2. Next, in Section D.2, we review two results needed to complete the proof of Proposition 3.6 concerning the continuity and differentiability of the dual objective.

3. Finally, in Section D.3 we leverage the preliminaries provided in Section D.2 to complete the proof of the proposition. This result is summarized in Proposition D.4.

Ultimately, the result in Proposition 4.4 is obtained by combining the results in Lemma D.1 and Proposition D.4 and using the union bound.

## D.1  Bounding the parametrization gap

In this section, we are interested in the relationship between the statistical problem (P-CON) and its dual problem. In particular, the dual problem to (P-CON) can be written in the following way

$$D^\star \triangleq \max_{\nu \geq 0} \min_{\boldsymbol{\theta} \in \Theta} L(\boldsymbol{\theta}, \nu) \tag{DI}$$

for the Lagrangian

$$L(\boldsymbol{\theta}, \nu) = \mathbb{E}\left[ \max_{\boldsymbol{\delta} \in \Delta} \ell\big(f_{\boldsymbol{\theta}}(\mathbf{x} + \boldsymbol{\delta}), y\big) \right] + \nu\left[ \mathbb{E}\left[\ell\big(f_{\boldsymbol{\theta}}(\mathbf{x}), y\big)\right] - \epsilon \right]. \tag{33}$$

The goal of this subsection is to prove the following lemma, which establishes bounds on the error induced by the parameterization space $\Theta$.

**Lemma D.1.** *Under the conditions of Prop. 4.4, the value $D^\star$ of* (DI) *is related to the value $P^\star$ of* (P-CON) *by*

$$P^\star - M\alpha \leq D^\star \leq P^\star. \tag{34}$$

*Proof.* The result in Lemma D.1 is trivial when the hypothesis class $\mathcal{H}$ induced by the parametrization is convex. In this case, (P-CON) is a convex program and Assumption 4.2 (Slater's condition) implies that it is strongly dual [96, Ch. 4]. In other words, $P^\star = D^\star$. Hence, we are interested in the setting in which $\mathcal{H}$ is not convex, but is still a rich parametrization as per (4.1) such as the class of CNNs.

In the nonconvex case, the upper bound in (34) is a simple consequence of weak duality [96, Ch. 4]. To obtain the lower bound, consider the variational problem

$$\tilde{P}^\star \triangleq \underset{\phi \in \overline{\mathcal{H}}}{\text{minimize}} \quad \mathbb{E}_{(\mathbf{x}, y) \sim \mathcal{D}}\left[ \max_{\boldsymbol{\delta} \in \Delta} \ell\big(\phi(\mathbf{x} + \boldsymbol{\delta}), y\big) \right] \tag{PVIII}$$
$$\text{subject to} \quad \mathbb{E}_{(\mathbf{x}, y) \sim \mathcal{D}}\left[ \ell\big(\phi(\mathbf{x}), y\big) \right] \leq \epsilon - M\alpha$$

for $\overline{\mathcal{H}} = \text{conv}(\mathcal{H})$. Let $\tilde{\phi}^\star \in \overline{\mathcal{H}}$ be a solution of (PVIII) associated with $\delta^\star(x, y)$, the perturbations that attains the maximum in its objective. Since $\Delta$ is a compact set by assumption, there indeed exists a perturbation $\delta^\star \in \Delta$ that achieves the maximum. Since $\overline{\mathcal{H}}$ is convex, (PVIII) is now a convex optimization problem (recall that the pointwise maximum of convex functions is convex [96, Prop 1.1.6]) which therefore has a strictly feasible point $f_{\boldsymbol{\theta}'} \in \mathcal{H} \subset \overline{\mathcal{H}}$ (Assumption 4.2). Hence, it is strongly dual [96, Ch. 4] and

$$\tilde{P}^\star = \max_{\tilde{\nu} \geq 0} \min_{\phi \in \overline{\mathcal{H}}} \tilde{L}(\phi, \tilde{\nu}) = \tilde{L}(\tilde{\phi}^\star, \tilde{\nu}^\star), \tag{35}$$

where $\tilde{\nu}^\star$ achieves the maximum in (35) for the Lagrangian[5]

$$\tilde{L}(\phi, \tilde{\nu}) = \mathbb{E}\left[ \max_{\boldsymbol{\delta} \in \Delta} \ell\big(\phi(\mathbf{x} + \boldsymbol{\delta}), y\big) \right] + \tilde{\nu}\left[ \mathbb{E}\left[\ell\big(\phi(\mathbf{x}), y\big)\right] - \epsilon + M\alpha \right] \tag{36}$$

---

[5]For clarity, we omit the distribution $\mathcal{D}$ over which the expectations are taken.

To proceed, notice from (DI) that

$$D^\star \geq \min_{\boldsymbol{\theta} \in \Theta} L(\boldsymbol{\theta}, \nu), \quad \text{for all } \nu \geq 0,$$

and that since $\mathcal{H} \subseteq \overline{\mathcal{H}} = \text{conv}(\mathcal{H})$, we obtain

$$D^\star \geq \min_{\boldsymbol{\theta} \in \Theta} L(\boldsymbol{\theta}, \tilde{\nu}^\star) \geq \min_{\phi \in \overline{\mathcal{H}}} \tilde{L}(\phi, \tilde{\nu}^\star). \tag{37}$$

Using the strong duality of (PVIII), the expression written above in (37) yields

$$D^\star \geq \min_{\phi \in \overline{\mathcal{H}}} \tilde{L}(\phi, \tilde{\nu}^\star) = \tilde{P}^\star = \mathbb{E}\Big[\ell\big(\phi^\star(\mathbf{x} + \delta^\star(\mathbf{x}, y)), y\big)\Big]. \tag{38}$$

Now note that to obtain the lower bound in (34), it suffices to show that

$$\mathbb{E}\Big[\ell\big(\phi^\star(\mathbf{x} + \delta^\star(\mathbf{x}, y)), y\big)\Big] \geq P^\star - M\alpha \tag{39}$$

To obtain this lower bound, notice from Assumption 4.1 that there exists $\tilde{\boldsymbol{\theta}}^\dagger \in \Theta$ such that

$$\sup_{\mathbf{x}} \left|\tilde{\phi}^\star(\mathbf{x}) - f_{\tilde{\boldsymbol{\theta}}^\dagger}(\mathbf{x})\right| \leq \alpha. \tag{40}$$

For these parameters, it holds that

$$\left|\mathbb{E}\left[\ell\big(\phi(\mathbf{x} + \delta^\star(\mathbf{x}, y)), y\big)\right] - \mathbb{E}\left[\max_{\boldsymbol{\delta} \in \Delta} \ell\big(f_{\tilde{\boldsymbol{\theta}}^\star}(\mathbf{x} + \boldsymbol{\delta}), y\big)\right]\right| \leq$$

$$\mathbb{E}\left[\left|\ell\big(\phi(\mathbf{x} + \delta^\star(\mathbf{x}, y)), y\big) - \max_{\boldsymbol{\delta} \in \Delta} \ell\big(f_{\tilde{\boldsymbol{\theta}}^\star}(\mathbf{x} + \boldsymbol{\delta}), y\big)\right|\right] \leq$$

$$\mathbb{E}\left[\left|\ell\big(\phi(\mathbf{x} + \delta^\star(\mathbf{x}, y)), y\big) - \ell\big(f_{\tilde{\boldsymbol{\theta}}^\star}(\mathbf{x} + \delta^\star(\mathbf{x}, y)), y\big)\right|\right],$$

where the first inequality is due to the convexity of the absolute value (Jensen's inequality) and the second inequality follows from the fact that $\delta^\star$ is a suboptimal solution of $\max_{\boldsymbol{\delta} \in \Delta} \ell\big(f_{\tilde{\boldsymbol{\theta}}^\star}(\mathbf{x} + \boldsymbol{\delta}), y\big)$. Using the Lipschitz continuity of the loss and Assumption 4.1, we conclude that

$$\left|\tilde{P}^\star - \mathbb{E}\left[\max_{\boldsymbol{\delta} \in \Delta} \ell\big(f_{\tilde{\boldsymbol{\theta}}^\star}(\mathbf{x} + \boldsymbol{\delta}), y\big)\right]\right| \leq M\,\mathbb{E}\left[\left|\phi(\mathbf{x} + \delta^\star(\mathbf{x}, y)) - f_{\tilde{\boldsymbol{\theta}}^\star}(\mathbf{x} + \delta^\star(\mathbf{x}, y))\right|\right] \leq M\alpha. \tag{41}$$

Using a similar argument, we also obtain that

$$\left|\mathbb{E}\left[\ell\big(\tilde{\phi}^\star(\mathbf{x}), y\big)\right] - \mathbb{E}\left[\ell\big(f_{\tilde{\boldsymbol{\theta}}^\star}(\mathbf{x}), y\big)\right]\right| \leq M\alpha. \tag{42}$$

Hence, given that $\tilde{\phi}^\star(\boldsymbol{x})$ is feasible for (PVIII), (42) implies that $\tilde{\boldsymbol{\theta}}^\star$ is feasible for (P-CON). By optimality, $P^\star \leq \mathbb{E}\left[\max_{\boldsymbol{\delta} \in \Delta} \ell\big(f_{\tilde{\boldsymbol{\theta}}^\star}(\mathbf{x} + \boldsymbol{\delta}), y\big)\right]$. Now recalling (38), we conclude that

$$D^\star \geq \mathbb{E}\Big[\ell\big(\phi^\star(\mathbf{x} + \delta^\star(\mathbf{x}, y)), y\big)\Big]$$

$$\geq P^\star + \mathbb{E}\Big[\ell\big(\phi^\star(\mathbf{x} + \delta^\star(\mathbf{x}, y)), y\big) - \max_{\boldsymbol{\delta} \in \Delta} \ell\big(f_{\tilde{\boldsymbol{\theta}}^\star}(\mathbf{x} + \boldsymbol{\delta}), y\big)\Big]$$

$$\geq P^\star - M\alpha,$$

where the last inequality follows from (41). □

### D.2 Preliminaries for proving the empirical gap

Before proceeding to considering the empirical gap of the dual problem, we state two preliminary results which will be useful in proving the empirical gap in Section D.3. To this end, we first state the classical result known as Danskin's theorem.

**Theorem D.2** (Danskin's Theorem). *Consider the function*

$$F(w) = \max_{z \in \mathcal{Z}} f(w, z) \tag{43}$$

*where $f : \mathbb{R}^n \times \mathcal{Z} \to \overline{\mathbb{R}}$ and assume that the following three conditions hold:*

*(i) $f(\cdot, z)$ is convex in $w$ for each $z \in \mathcal{Z}$;*

*(ii) $f(w, \cdot)$ is continuous in $z$ for each $w$ in a certain neighborhood of a point $w_0$;*

*(iii) The set $\mathcal{Z}$ is compact.*

*Then it holds that*

$$\partial F(x_0) = \mathrm{conv}\left(\bigcup_{z \in \hat{\mathcal{Z}}(w_0)} \partial_w f(w_0, z))\right) \tag{44}$$

*where $\hat{\mathcal{Z}}(w)$ denotes the set of $z \in \mathcal{Z}$ at which $F(w) = f(w, z)$.*

The interested reader can find a full proof of Danskin's theorem in [98, Thm. 2.87].

Next, we note that in the proof presented in Section D.3, it will be necessary to verify that a function analogous to $F(w)$ defined in (43) which is defined as the pointwise maximum of continuous function $f(w, z)$ is continuous. To verify this continuity, we will rely on the following result:

**Lemma D.3.** *Fix any point $u_0 \in \Phi$ and let $g : \Phi \times \mathbb{R}^n \to \overline{\mathbb{R}}$. Denote*

$$v(u) = \inf_{w \in \Phi(u)} g(w, u) \quad and \quad \mathcal{S}(u) = \underset{w \in \Phi(u)}{\mathrm{argmin}}\, g(w, u) \tag{45}$$

*Now suppose that*

*(a) The function $g(w, u)$ is continuous on $\Phi \times \mathbb{R}^n$;*

*(b) The feasible set $\Phi$ is closed;*

*(c) There exists a constant $\alpha \in \mathbb{R}$ and a compact set $C \subset \mathbb{R}^n$ such that for every $u$ in a neighborhood of $u_0$, the level set $\{w \in \Phi(u) : g(w, u) \leq \alpha\}$ is nonempty and contained in $C$;*

*(d) For any neighborhood $\mathcal{N}$ of $\mathcal{S}(u_0)$, there exists a neighborhood $\mathcal{N}_U$ of $u_0$ such that $\mathcal{N} \cap \Phi(u) \neq \varnothing \,\forall u \in \mathcal{N}_U$.*

*Then it holds that $v(u)$ is continuous at $u = u_0$.*

Further details regarding this result as well as a full proof can be found in [99, Prop. 4.4].

### D.3 Bounding the empirical gap

We now proceed by evaluating the empirical gap between the statistical dual problem (DI) and its empirical version $(\widehat{\mathrm{DI}})$. In particular, our goal is to prove the following result.

**Proposition D.4.** *Let $\hat{\nu}$ be a solution of $(\widehat{\mathrm{DI}})$ with a finite $D^\star$. Under the conditions of Theorem 4.4, there exists $\hat{\boldsymbol{\theta}}^\star \in \mathrm{argmin}_{\boldsymbol{\theta} \in \Theta}\, \hat{L}(\boldsymbol{\theta}, \hat{\nu}^\star)$ such that*

$$\left|D^\star - \hat{D}^\star\right| \leq (1 + \overline{\nu}) \max(\zeta_R(N), \zeta_N(N)) \quad \text{(near-optimality)} \tag{46}$$

$$\mathbb{E}_{(\boldsymbol{x},y) \sim \mathcal{D}}\left[\ell_i\big(f_{\hat{\boldsymbol{\theta}}^\star}(\boldsymbol{x}), y\big)\right] \leq c_i + \zeta_i(N_i) \quad \text{(near-feasibility)}. \tag{47}$$

*hold with probability $1 - 5\delta$, where $\overline{\nu} = \max(\hat{\nu}^\star, \nu^\star)$, for $\hat{\nu}^\star$ a solution of $(\widehat{\mathrm{DI}})$ and $\nu^\star$ a solution of (DI), and $D^\star$ and $\hat{D}^\star$ as in (DI) and $(\widehat{\mathrm{DI}})$ respectively.*

*Proof.* **(Near-optimality).** Let $\nu^\star$ and $\hat{\nu}^\star$ be solutions of (DI) and $(\widehat{\mathrm{DI}})$ respectively and consider the set of dual minimizers

$$\Theta^\dagger(\nu) = \underset{\theta \in \Theta}{\mathrm{argmin}}\, L(\boldsymbol{\theta}, \nu) \quad \text{and} \quad \hat{\Theta}^\dagger(\hat{\nu}) = \underset{\theta \in \Theta}{\mathrm{argmin}}\, \hat{L}(\boldsymbol{\theta}, \hat{\nu})$$

Using the optimality of $\nu^\star$, it holds that

$$D^\star - \hat{D}^\star = \min_{\boldsymbol{\theta} \in \Theta} L(\boldsymbol{\theta}, \nu^\star) - \min_{\boldsymbol{\theta} \in \Theta} \hat{L}(\boldsymbol{\theta}, \hat{\nu}^\star)$$

$$\leq \min_{\boldsymbol{\theta} \in \Theta} L(\boldsymbol{\theta}, \nu^\star) - \min_{\boldsymbol{\theta} \in \Theta} \hat{L}(\boldsymbol{\theta}, \nu^\star).$$

Since $\hat{\boldsymbol{\theta}}^\dagger \in \hat{\Theta}^\dagger(\nu^\star)$ is suboptimal for $L(\boldsymbol{\theta}, \nu^\star)$, we get

$$D^\star - \hat{D}^\star \leq L(\hat{\boldsymbol{\theta}}^\dagger, \nu^\star) - \hat{L}(\hat{\boldsymbol{\theta}}^\dagger, \nu^\star). \tag{48}$$

Using a similar argument yields

$$D^\star - \hat{D}^\star \geq L(\boldsymbol{\theta}^\dagger, \hat{\nu}^\star) - \hat{L}(\boldsymbol{\theta}^\dagger, \hat{\nu}^\star) \tag{49}$$

for $\boldsymbol{\theta}^\dagger \in \Theta^\dagger(\hat{\nu}^\star)$. Thus, we obtain that

$$\left| D^\star - \hat{D}^\star \right| \leq \max\left\{ \left| L(\hat{\boldsymbol{\theta}}^\dagger, \nu^\star) - \hat{L}(\hat{\boldsymbol{\theta}}^\dagger, \nu^\star) \right|, \left| L(\boldsymbol{\theta}^\dagger, \hat{\nu}^\star) - \hat{L}(\boldsymbol{\theta}^\dagger, \hat{\nu}^\star) \right| \right\} \tag{50}$$

Using the empirical bound from Assumption 4.3, we obtain that

$$\left| L(\boldsymbol{\theta}, \nu) - \hat{L}(\boldsymbol{\theta}, \nu) \right| \leq \zeta_R(N) + \nu\zeta_N(N), \tag{51}$$

holds uniformly over $\boldsymbol{\theta}$ with probability $1 - 4\delta$.

**(Near-feasibility).** The proof relies on characterizing the superdifferential of the dual function

$$\hat{d}(\nu) = \min_{\boldsymbol{\theta}\in\Theta} \hat{L}(\boldsymbol{\theta}, \nu) \tag{52}$$

from $(\widehat{\mathrm{DI}})$. Explicitly, we say $p \in \mathbb{R}$ is a *supergradient* of $\hat{d}$ at $\nu$ if

$$\hat{d}(\nu') \leq \hat{d}(\nu) + p(\nu' - \nu), \text{ for all } \nu' \geq 0. \tag{53}$$

The set of all supergradients of $\hat{d}$ at $\nu$ is called the *superdifferential* of $\hat{d}$ at $\nu$ and is denoted $\partial\hat{d}(\nu)$. To characterize the supperdifferential, first let $\Theta^\dagger(\nu) \in \mathrm{argmin}_{\boldsymbol{\theta}\in\Theta} \hat{L}(\boldsymbol{\theta}, \nu)$ for the Lagrangian $\hat{L}$ and define the constraint slack as

$$s(\boldsymbol{\theta}) = \left[ \frac{1}{N} \sum_{n=1}^{N} \ell\big(f_{\boldsymbol{\theta}}(\mathbf{x}_n), y_n\big) - \epsilon \right]_+. \tag{54}$$

Now by rewriting $\hat{d}(\nu)$ as follows

$$\hat{d}(\nu) = \min_{\boldsymbol{\theta}\in\Theta} \hat{L}(\boldsymbol{\theta}, \nu) = -\max_{\boldsymbol{\theta}\in\Theta} -\hat{L}(\boldsymbol{\theta}, \nu), \tag{55}$$

we argue that the conditions of Thm. D.2 are satisfied. To begin, we note that because $\hat{L}(\boldsymbol{\theta}, \cdot)$ is affine in $\nu$ for all $\boldsymbol{\theta} \in \Theta$ and $\Theta$ is compact, we immediately meet conditions (i) and (iii) of Thm. D.2. Thus, it suffices to show that $\hat{L}(\cdot, \nu)$ is continuous in $\boldsymbol{\theta}$ for all $\nu \geq 0$.

To prove the continuity of $\hat{d}(\nu)$, we will seek to show that the conditions of Lemma D.3 are satisfied for $\hat{d}(\nu)$. In this way, first recall that $\ell(\cdot, y)$ is continuous for all $y \in \mathcal{Y}$ and $f_{\boldsymbol{\theta}}(\mathbf{x})$ is differentiable with respect to $\boldsymbol{\theta}$ and $\mathbf{x}$. Therefore, it holds that $\ell(f_{\boldsymbol{\theta}}(\mathbf{x}), y)$ is continuous on $\Omega \times \Theta$. Thus, property (a) in Lemma D.3 holds. Furthermore, the fact that $\Delta$ is compact and fixed with respect to $\boldsymbol{\theta}$ establishes (b) and (d). Finally, condition (c) holds by observing that the perturbation $\boldsymbol{\delta}$ are finite dimensional and that $\ell(f_{\boldsymbol{\theta}}(\mathbf{x} + \boldsymbol{\delta}), y)$ is uniformly bounded.

Having established the continuity of $\hat{d}(\nu)$, altogether we have shown that the conditions (i)–(iii) of Thm. D.2 are satisfied. Thus, using the fact that $\hat{d}(\nu) = \min_{\boldsymbol{\theta}\in\Theta} \hat{L}(\boldsymbol{\theta}, \nu) = -\max_{\boldsymbol{\theta}\in\Theta} -\hat{L}(\boldsymbol{\theta}, \nu)$, Thm. D.2 yields the following result:

$$\partial d(\boldsymbol{\mu}) = \mathrm{conv}\left( \bigcup_{\boldsymbol{\theta}^\dagger\in\Theta^\dagger(\boldsymbol{\mu})} s(\boldsymbol{\theta}^\dagger) \right). \tag{56}$$

To complete the proof, we assume toward contradiction that for all $\hat{\boldsymbol{\theta}}^\dagger \in \hat{\Theta}^\dagger(\hat{\nu}^\star)$ it holds that

$$\frac{1}{N} \sum_{n=1}^{N} \ell\big(f_{\hat{\boldsymbol{\theta}}^\dagger}(\mathbf{x}_n), y_n\big) > \epsilon.$$

Then, from our discussion of the superdifferential, $\mathbf{0} \notin \partial d(\hat{\nu}^\star)$, which contradicts the optimality of $\hat{\nu}^\star$. Hence, there must be $\hat{\boldsymbol{\theta}}^\dagger \in \hat{\Theta}^\dagger(\hat{\boldsymbol{\mu}}^\star)$ such that $\frac{1}{N} \sum_{n=1}^N \ell\big(f_{\hat{\boldsymbol{\theta}}^\dagger}(\mathbf{x}_n), y_n\big) \leq \epsilon$. For those parameters, the uniform bound in Assumption 4.3, yields that, with probability $1 - \delta$ over the data,

$$\mathbb{E}_{(\boldsymbol{x},y) \sim \mathcal{D}}\Big[\ell\big(f_{\hat{\boldsymbol{\theta}}^\dagger}(\mathbf{x}), y\big)\Big] \leq \frac{1}{N} \sum_{n=1}^N \ell\big(f_{\hat{\boldsymbol{\theta}}^\dagger}(\boldsymbol{x}_n), y_n\big) + \zeta_N(N) \leq \epsilon + \zeta_N(N). \tag{57}$$

Combining (50), (51), and (57) using the union bound concludes the proof. $\qquad \square$

## E  Deriving the Langevin Monte Carlo sampler

In this appendix, we offer a more detailed derivation of the Langevin Monte Carlo sampler used in Algorithm 1. Along the way, we present a brief, expository introduction to Hamiltonian Monte Carlo to provide the reader with further context concerning the derivation of Algorithm 1. Much of this material is based on the derivations provided in standard references, including [100, 101, 94]; we refer the reader to these references for a more complete treatment of these topics.

In the setting of our paper, given a fixed data point $(\mathbf{x}, y) \in \Omega$, our goal in deriving the sampler is to evaluate the following expectation:

$$\mathbb{E}_{\boldsymbol{\delta} \sim \lambda(\boldsymbol{\delta}|\mathbf{x}, y)} \left[ \ell(f_{\boldsymbol{\theta}}(\mathbf{x} + \boldsymbol{\delta}), y) \right] = \int_{\Delta} \ell(f_{\boldsymbol{\theta}}(\mathbf{x} + \boldsymbol{\delta}), y) \lambda(\boldsymbol{\delta}|\mathbf{x}, y) \tag{58}$$

where $\lambda$ denotes the perturbation distribution defined by

$$\lambda(\boldsymbol{\delta}|\mathbf{x}, y) = \frac{\ell(f_{\boldsymbol{\theta}}(\mathbf{x} + \boldsymbol{\delta}), y)}{\int_{\Delta} \ell(f_{\boldsymbol{\theta}}(\mathbf{x} + \boldsymbol{\delta}), y) d\boldsymbol{\delta}} \tag{59}$$

and where $f_{\boldsymbol{\theta}} \in \mathcal{F}$ is a fixed classifier. Roughly speaking, this problem is challenging due to the fact that we cannot compute the normalization constant in (59). Therefore, although the form of (59) indicates that the amount of mass placed on $\boldsymbol{\delta} \in \Delta$ will be proportional to the loss $\ell(f_{\boldsymbol{\theta}}(\mathbf{x}, y))$ when the data is perturbed by this perturbation $\boldsymbol{\delta}$, it's unclear how we can sample from this distribution in practice.

The first step in deriving the sampler is to introduce a *momentum* variable $\boldsymbol{p}$ to complement the space of perturbations: $\boldsymbol{\delta} \to (\boldsymbol{\delta}, \boldsymbol{p})$. This transformation expands the $d$-dimensional perturbation space to a $2d$-dimensional *phase space*. Furthermore, this augmentation facilitates the lifting of $\lambda$ onto the so-called *canonical distribution* $\boldsymbol{\lambda}(\boldsymbol{\delta}, \boldsymbol{p})$ defined by

$$\boldsymbol{\lambda}(\boldsymbol{\delta}, \boldsymbol{p}) = \boldsymbol{\lambda}(\boldsymbol{p}|\boldsymbol{\delta}) \cdot \boldsymbol{\lambda}(\boldsymbol{\delta}), \tag{60}$$

which takes support over the $2d$-dimensional phase space. Notably, as we have artificially introduced the momentum parameters $\boldsymbol{p}$, the canonical density does not depend on a particular choice of parameterization of (60), and we can therefore express it in terms of an invariant *Hamiltonian function* $H(\boldsymbol{\delta}, \boldsymbol{p})$ defined by

$$\boldsymbol{\lambda}(\boldsymbol{\delta}, \boldsymbol{p}) = \exp\left\{ -H(\boldsymbol{\delta}, \boldsymbol{p}) \right\}, \quad \text{or equivalently} \quad H(\boldsymbol{\delta}, \boldsymbol{p}) = -\log \boldsymbol{\lambda}(\boldsymbol{\delta}, \boldsymbol{p}). \tag{61}$$

Now, owing to the decomposition in (60), note that the Hamiltonian can be written as follows:

$$H(\boldsymbol{\delta}, \boldsymbol{p}) = -\log \boldsymbol{\lambda}(\boldsymbol{p}|\boldsymbol{\delta}) - \log \boldsymbol{\lambda}(\boldsymbol{\delta}) \tag{62}$$

$$\equiv K(\boldsymbol{p}) + U(\boldsymbol{\delta}). \tag{63}$$

where we have defined a *kinetic energy* term $K(\boldsymbol{p}) = -\log \boldsymbol{\lambda}(\boldsymbol{p}|\boldsymbol{\delta})$ as well as a *potential energy* term $U(\boldsymbol{\delta}) = -\log \boldsymbol{\lambda}(\boldsymbol{\delta})$. By evolving the parameters $(\boldsymbol{\delta}, \boldsymbol{p})$ in phase space according to Hamilton's equations

$$\frac{d\boldsymbol{\delta}}{dt} = +\frac{\partial H}{\partial \boldsymbol{p}} = \frac{\partial K}{\partial \boldsymbol{p}} \quad \text{and} \quad \frac{d\boldsymbol{p}}{dt} = -\frac{\partial H}{\partial \boldsymbol{\delta}} = -\frac{\partial U}{\partial \boldsymbol{\delta}}, \tag{64}$$

we generate a trajectory $\boldsymbol{\delta}(t)$ that walks along the so-called *typical set* of the perturbation distribution $\boldsymbol{\lambda}(\boldsymbol{\delta})$ from which we want to sample. Thus, to generate such a trajectory, we first choose a distribution for the momentum parameters $\boldsymbol{p}$ and the integrate Hamilton's equations over time.

As is common in the literature, the next step in deriving the sampler is to place a probabilistic prior over $\boldsymbol{p}$. In what follows, we describe the samplers that result from two different priors.

**Gaussian prior.** As is common in the literature, one can take the prior over $\boldsymbol{p}$ to be a normal. That is, we can let $\boldsymbol{p} \sim \mathcal{N}(0, T I_d)$ where $T > 0$ is a constant and $I_d$ is the $d$-dimensional identity matrix. This in turn engenders a kinetic energy term $K(\boldsymbol{\delta}, \boldsymbol{p}) \propto (2T)^{-1} \|\boldsymbol{p}\|_2^2$. Then, to (approximately) integrate Hamilton's equations, we employ the following *leapfrog integration* update scheme:

$$\boldsymbol{\delta} \leftarrow \boldsymbol{\delta} + \eta \nabla_{\boldsymbol{\delta}} U(\boldsymbol{\delta}) + \sqrt{2\eta T} \boldsymbol{p} \quad \text{where} \quad \boldsymbol{p} \sim \mathcal{N}(0, T I_d). \tag{65}$$

**Laplacian prior.** Another common choice is to take the prior over $p$ to be Laplacian so that $p \sim \text{Laplacian}(0, T^2)$. A similar calculation to the one performed for the Gaussian prior reveals that this implies that $K(\delta, p) \propto \frac{1}{T} \|p\|_1$. Integrating Hamiltonian's equations for this choice of the kinetic energy function yields the following scheme:

$$\delta \leftarrow \delta + \eta \, \text{sign} \left[ \nabla_{\delta} U(\delta) + \sqrt{2\eta T} \xi \right] \quad \text{where} \quad \xi \sim \text{Laplace}(0, T). \tag{66}$$

Table 3: **Public implementations of baseline methods.** In this table, we list the public implementations of popular adversarial training methods that we used to train baseline classifiers.

| Algorithm | Implementation |
|-----------|----------------|
| PGD | `https://github.com/MadryLab/robustness` |
| TRADES | `https://github.com/yaodongyu/TRADES` |
| MART | `https://github.com/YisenWang/MART` |

## F   Further experimental details

In this appendix, we provide further experimental details beyond those given in the main text. All experiments were run across twelve NVIDIA RTX 5000 GPUs.

**Training hyperparameters and data loading.**   We record the hyperparameters used for training the neural networks in Section 6 on MNIST and CIFAR-10 below.

- **MNIST.** We train CNNs with two convolutional layers and two feed-forward layers. In particular, we use the architecture from the MNIST PyTorch tutorial; the full architecture is described in the following file: `https://github.com/pytorch/examples/blob/master/mnist/main.py`. We use a batch size of 128. All adversarial perturbations are defined over the perturbation set $\Delta = \{\boldsymbol{\delta} \in \mathbb{R}^d : \|\boldsymbol{\delta}\|_\infty \leq 0.3\}$. All models were trained for 50 epochs with the Adadelta optimizer, and we used a learning rate of 1.0.

- **CIFAR-10.** We train ResNet-18 and ResNet-50 classifiers with SGD and an initial learning rate of 0.01. We use 0.9 for the momentum, and we use weight decay with a penalty weight of $3.5 \times 10^{-3}$. We train all classifiers for 200 epochs, and we decay the learning rate by a factor of 10 at epochs 150, 175, and 190. In general, this is longer than CIFAR-10 classifiers are generally trained. We increased the number of epochs to allow the dual variable to converge before the first learning rate step. For completeness, we ran all baselines using the more standard training scheme of 120 total epochs with learning rate decays after epochs 55, 75, and 90; we noticed almost no difference in the final performance of the baselines for this shorter schedule. We also apply random crops and random horizontal flips to the training data. All adversarial perturbations are defined over the perturbation set $\Delta = \{\boldsymbol{\delta} \in \mathbb{R}^d : \|\boldsymbol{\delta}\|_\infty \leq 8/255\}$.

**Baseline implementations.**   As mentioned in the main text, we reran all baselines by adapting implementations released in prior work. In particular, our implementations of the baseline methods are based on the public implementations recorded in Table 3. These methods are all implemented in our repository, which is publicly available at the following link: `https://github.com/arobey1/advbench`.

**Baseline hyperparameters.**   Throughout the experiments section, we trained numerous baseline classifiers to offer points of comparison to our methods. In this section, we list the hyperparameters used for each of these methods:

- **PGD.** On MNIST, we used 7 projected gradient ascent steps with a step size of 0.1. On CIFAR-10, unless otherwise stated, we used 10 projected gradient ascent steps with a step size of 2/255. Note that in Table 7, we varied the number of ascent steps for PGD.

- **CLP & ALP.** The same step sizes and number of ascent steps were used for CLP and ALP as we reported above for PGD. In line with [42], we used a trade-off weight of $\lambda = 1.0$ for both methods on MNIST and CIFAR-10.

- **TRADES.** The same step sizes and number of ascent steps were used for TRADES as we reported above for PGD. Following [53], we used a trade-of weight of $\beta = 1/\lambda = 6.0$ for both datasets.

- **MART.** The same step sizes and number of ascent steps were used for MART as we reported above for PGD. Following [54], we used a trade-off weight of $\lambda = 5.0$ for both datasets.

**DALE hyperparameters.** Unlike methods many of the baselines described above, DALE does not require the user to manually tune a weight which controls the trade-off between multiple objectives. Instead, we use a primal-dual scheme to dynamically and adaptively update the weight on the clean objective. Below, we provide some discussion of the hyperparameters inherent to our primal-dual approach.

- **Margin $\rho$.** For MNIST, we found that a margin of 0.1 yielded strong performance.

- **Dual step size $\eta_d$.** We found that the dual step size should be chosen to be significantly smaller than the primal step size. By sweeping over $\eta_d \in \{0.1, 0.05, 0.01, 0.005, 0.005, 0.0001, 0.0005\}$, we found that a dual step size of $\eta_d = 0.001$ worked well in practice for CIFAR-10.

- **Primal step size $\eta_p$.** As described at the beginning of this appendix, we used $\eta_p = 1.0$ for MNIST and $\eta_p = 0.01$ for CIFAR-10.

- **Temperature T.** In practice, we found that the temperature should be chosen so that the noise coefficient $\sqrt{2\eta T}$ is relatively small. By sweeping over $\sqrt{2\eta T} \in \{10^{-1}, 10^{-2}, 10^{-3}, 10^{-4}, 10^{-5}, 10^{-6}\}$, we found that robust performance began to degrade for $\sqrt{2\eta T} > 10^{-4}$. For MNIST, we found that $\sqrt{2\eta T} = 10^{-3}$ worked well; on CIFAR-10, we used $\sqrt{2\eta T} = 10^{-4}$.

Table 4: **Accuracy and computational complexity on MNIST.** In this table, we report the test accuracy and computational complexity of our method and various state-of-the-art baselines on MNIST. In particular, all methods are trained using a four-layer CNN architecture, and we use a perturbation set of $\Delta = \{\boldsymbol{\delta} \in \mathbb{R}^d : \|\boldsymbol{\delta}\|_\infty \leq 0.3\}$. Our results are highlighted in gray.

| Algorithm | $\rho$ | Test accuracy (%) | | | Performance (sec.) | |
| | | Clean | FGSM | PGD$^{10}$ | Batch | Epoch |
|---|---|---|---|---|---|---|
| ERM | - | 99.3 | 14.3 | 1.46 | 0.007 | 3.47 |
| FGSM | - | 98.3 | 98.1 | 13.0 | 0.011 | 5.48 |
| PGD | - | 98.1 | 95.5 | 93.1 | 0.039 | 18.2 |
| CLP | - | 98.0 | 95.4 | 92.2 | 0.047 | 21.9 |
| ALP | - | 98.1 | 95.5 | 92.5 | 0.048 | 22.0 |
| TRADES | - | 98.9 | 96.5 | 94.0 | 0.055 | 25.8 |
| MART | - | 98.9 | 96.1 | 93.5 | 0.043 | 20.4 |
| DALE | 1.0 | 99.1 | 97.7 | 94.5 | 0.053 | 25.4 |

# G  Further experiments

## G.1  MNIST

We first consider the MNIST dataset [102]. All models use a four-layer CNN architecture trained using the Adadelta optimizer [103]. To evaluate the robust performance of trained models, we report the test accuracy with respect to two independent adversaries. In particular, we use a 1-step and a 10-step PGD adversary to evaluate robust performance; we denote these adversaries by FGSM and PGD$^{10}$ respectively.

A summary of the performance of DALE and various state-of-the-art baselines is shown in Table 4. Notice that DALE marginally outperforms each of the baselines in robust accuracy, while maintaining a clean accuracy that is similar to that of ERM. This indicates that on MNIST, DALE is able to reach high robust accuracies without trading off in nominal performance. This table also shows a runtime analysis of each of the methods. Notably, DALE and TRADES have similar running times, which is likely due to the fact in our implementation of DALE, we use the same KL-divergence loss to search for challenging perturbations.

## G.2  CIFAR-10

We next consider the CIFAR10 dataset [104]. Throughout this section, we use the ResNet-18 architecture trained using SGD, and we consider adversaries which can can generate perturbations $\boldsymbol{\delta}$ lying within the perturbation set $\Delta = \{\boldsymbol{\delta} \in \mathbb{R}^d : \|\boldsymbol{\delta}\|_\infty \leq 8/255\}$. To this end, we use evaluate the robust performance of trained models using FGSM and PGD$^{20}$ adversaries. For all of the classifiers trained using DALE, we use the KL-divergence loss for $\ell_{\text{pert}}$ and $\ell_{\text{ro}}$, and we use the cross-entropy loss for $\ell_{\text{nom}}$.

In Table 5, we show a summary of our results on CIFAR-10. One notable aspect of our results is that DALE trained with $\rho = 0.8$ is the only model to achieve greater than 85% clean accuracy and greater than 50% robust accuracy. This indicates that DALE is more successfully able to mitigate the trade-off between robustness and nominal performance. And indeed, the baselines that have relatively high robust accuracy (TRADES and MART) suffer a significant drop in clean accuracy relative to DALE (-4.3% for TRADES and -6.1% for MART when compared with DALE trained with $\rho = 0.8$). Table 5 also shows a comparison of the computation time for each of the methods. These results indicate that the computational complexity of DALE is on a par with TRADES.

**A closer look at the trade-off between accuracy and robustness.**   We next study the trade-off between robustness and nominal performance of DALE for two separate architectures: ResNet-18 and ResNet-50. In our formulation, the parameter $\rho$ explicitly captures this trade-off in the sense that a smaller $\rho$ will require a higher level of nominal performance, which in turn reduces the size of the feasible set. This reduction has the effect of limiting the robust performance of the classifier.

Table 5: **Accuracy and computational complexity on CIFAR-10.** In this table, we report the test accuracy and computational complexity of our method and various state-of-the-art baselines on CIFAR-10. In particular, all methods are trained using a ResNet-18 architecture, and we use a perturbation set of $\Delta = \{\boldsymbol{\delta} \in \mathbb{R}^d : \|\boldsymbol{\delta}\|_\infty \leq 8/255\}$. Our results are highlighted in gray. Of note is the fact that our method advances the state-of-the-art both in adversarial and in clean accuracy.

| | | Test accuracy (%) | | | Performance (sec.) | |
|---|---|---|---|---|---|---|
| **Algorithm** | $\rho$ | **Clean** | **FGSM** | **PGD$^{20}$** | **Batch** | **Epoch** |
| ERM | - | 94.0 | 0.01 | 0.01 | 0.073 | 28.1 |
| FGSM | - | 72.6 | 49.7 | 40.7 | 0.135 | 53.0 |
| PGD | - | 83.8 | 53.7 | 48.1 | 0.735 | 287.9 |
| CLP | - | 79.8 | 53.9 | 48.4 | 0.872 | 340.5 |
| ALP | - | 75.9 | 55.0 | 48.8 | 0.873 | 341.2 |
| TRADES | - | 80.7 | 55.2 | 49.6 | 1.081 | 422.0 |
| MART | - | 78.9 | 55.6 | 49.8 | 0.805 | 314.1 |
| DALE | 0.5 | **86.0** | 54.4 | 48.4 | 1.097 | 421.4 |
| DALE | 0.8 | 85.0 | 55.4 | 50.1 | 1.098 | 422.6 |
| DALE | 1.1 | 82.1 | 55.2 | **51.7** | 1.097 | 421.0 |

Table 6: **Evaluating the trade-off between robustness and accuracy.** To evaluate the trade-off between robustness and nominal performance, we train ResNet-18 and ResNet-50 models on CIFAR-10 for different trade-off parameters $\rho$. Notice that across both architectures, the impact of increasing $\rho$ is to simultaneously decrease clean performance and increase robust performance.

| | ResNet-18 | | | Resnet-50 | | |
|---|---|---|---|---|---|---|
| $\rho$ | **Clean** | **FGSM** | **PGD$^{20}$** | **Clean** | **FGSM** | **PGD$^{20}$** |
| 0.1 | 93.0 | 35.6 | 1.50 | 93.8 | 23.9 | 16.7 |
| 0.2 | 92.4 | 43.6 | 11.9 | 93.7 | 20.5 | 16.3 |
| 0.3 | 88.7 | 42.4 | 31.2 | 90.1 | 43.0 | 24.8 |
| 0.4 | 86.4 | 50.9 | 44.3 | 86.2 | 50.5 | 38.4 |
| 0.5 | 86.0 | 54.4 | 48.4 | 86.5 | 50.1 | 42.6 |
| 0.6 | 85.6 | 54.6 | 49.0 | 86.1 | 57.7 | 52.0 |
| 0.7 | 85.3 | 56.2 | 50.3 | 84.7 | 57.0 | 51.4 |
| 0.8 | 83.8 | 55.4 | 50.1 | 84.3 | 56.4 | 50.8 |
| 0.9 | 83.8 | 56.0 | 51.3 | 83.9 | 55.9 | 51.2 |
| 1.0 | 82.2 | 54.7 | 51.2 | 82.1 | 54.2 | 50.1 |
| 1.1 | 82.1 | 55.2 | 51.7 | 80.4 | 52.3 | 49.9 |

In Table 6, we illustrate this trade-off by varying $\rho$ from 0.1 to 1.1. For both architectures, the trade-off is clearly reflected in the fact that increasing the margin $\rho$ has the simultaneous effect of decreasing the clean accuracy and increasing the robust accuracy for both adversaries. We highlight that for the ResNet-18 architecture, when the constraint is enforced with a relatively large margin (e.g., $\rho \geq 1.0$), DALE achieves nearly 52% robust accuracy against PGD$^{20}$, which is nearly two percentage points higher than any of the baseline classifiers in Table 1. On the other hand, when the margin is relatively small (e.g., $\rho \leq 0.2$), there is almost no trade-off in the clean accuracy relative to ERM in Table 1, although as a result of this small margin, the robust performance takes a significant hit. Interestingly, with regard to the classifiers trained using ResNet-50, it seems to be the case that the margin $\rho$ corresponding to the largest robust accuracy is different than the peak for ResNet-18.

**Impact of the number of Langevin iterations.** In Table 7, we study the impact of varying the number of Langevin iterations $L$. For each row in this table, we train a ResNet-18 classifier with $\rho = 1.0$; as before, we use the KL-divergence loss for $\ell_{\text{pert}}$ and $\ell_{\text{ro}}$, and we use the cross-entropy loss for $\ell_{\text{nom}}$. As one would expect, when $L$ is small, the trained classifiers have relatively high clean

Table 7: **Impact of the number of ascent steps.** In this table, we show the impact of varying the number of Langevin steps used by DALE in lines 4-7 of Algorithm 1. To offer a point of comparison, we also show the impact of varying the number of ascent steps for PGD.

| | PGD$^L$ | | | DALE | | |
|---|---|---|---|---|---|---|
| **L** | **Clean** | **FGSM** | **PGD$^{20}$** | **Clean** | **FGSM** | **PGD$^{20}$** |
| 1 | 92.9 | 52.3 | 23.7 | 87.2 | 46.6 | 39.0 |
| 2 | 90.9 | 49.9 | 36.6 | 85.4 | 53.6 | 47.1 |
| 3 | 87.7 | 50.6 | 41.5 | 84.0 | 55.0 | 50.2 |
| 4 | 84.5 | 52.2 | 43.3 | 82.8 | 55.0 | 50.7 |
| 5 | 83.6 | 53.5 | 47.9 | 82.5 | 54.9 | 50.7 |
| 10 | 83.8 | 53.7 | 48.1 | 82.2 | 54.7 | 51.2 |
| 15 | 82.9 | 54.0 | 48.0 | 81.0 | 54.7 | 51.0 |
| 20 | 83.0 | 54.4 | 48.3 | 81.0 | 54.7 | 51.4 |

accuracy and relatively low robust accuracy. To this end, increasing $L$ has the simultaneous effect of decreasing clean accuracy and increasing robust accuracy.

To offer a point of comparison, we also show the analogous results for PGD run using the cross-entropy loss, where $L$ is taken to be the number of steps of projected gradient ascent. As each Langevin iteration of DALE effectively amounts to a step of projected gradient ascent with noise, we expect that the impact of varying $L$ in DALE will be analogous to the impact of varying the number of training-time PGD steps. And indeed, as we increase $L$, the robust performance of PGD improves and the clean performance decreases.

# H   On the convergence of Algorithm 1

Observe that Algorithm 1 is a primal-dual algorithm [105] in which the sampling procedure in steps 3–7 is used to obtain an estimate of the stochastic gradient of the primal problem. When $\theta \mapsto \ell\big(f_{\theta}(\cdot), \cdot\big)$ is convex (e.g., for linear, kernel, or logistic models), it is well-known that SGD converges almost surely as long as this gradient estimate is unbiased [106]. As is typical with LMC, we omitted the Metropolis-Hastings acceptance step in Algorithm 1 that would guarantee unbiased estimates [94]. Still, when $g$ is log-concave (e.g., the softmax output of a CNN), this procedure approaches the true distribution in total variation norm, which implies that its bias can be made arbitrarily small [81]. This is enough to guarantee almost sure convergence to a neighborhood of the optimum [107, 108].

The convergence properties of primal-dual methods are less well understood when $\theta \mapsto \ell\big(f_{\theta}(\cdot), \cdot\big)$ is non-convex. Nevertheless, a good estimate of the primal minimizer is enough to obtain an approximate gradient for dual ascent [59, 60]. There is overwhelming empirical and theoretical evidence that this is the case for overparametrized models, such as CNNs, trained using gradient descent [35–39]. We can then run the primal (step 8) and dual (step 10) updates at different timescales so as to obtain a good estimate of the primal minimizer before performing dual ascent.

# I Further related work

**Adversarial robustness.** As described in Section 1, it is well-know that state-of-the-art classifiers are susceptible to adversarial attacks [11–17, 26]. Toward addressing this challenging, a rapidly-growing body of work has provided *attack algorithms* to generate data perturbations that fool classifiers and *defense algorithms* which are designed to train robust classifiers to be robust against these perturbations. However, despite the myriad of work in this field and significant improvements on a number of well-known benchmarks [109–114, 27–31], there are still many open questions on when adversarial learning is even possible and in what sense [67–71]. Unlike the majority of these works, we exploit duality to derive a principled primal-dual style algorithm from first principles for the adversarial robustness setting.

**Constrained optimization.** Also related are works that seek to enforce constraints on learning problems [115]. While several heuristic algorithms exist for this setting, many focus on restricted classes of constraints [116–120] and those that can handle more general constraints come at the cost of added computation complexity [121, 122]. Moreover, each of these works seeks to enforce constraints on a particular parameterization for the learning problem (such as directly on the weights of a neural network) rather than on the underlying statistical problem, as we do in this paper. In this way, our work is more related to the primal-dual style algorithms which often arise in convex optimization [105, 77].