# OpenReview forum: "Adversarial Robustness with Semi-Infinite Constrained Learning"
_NeurIPS.cc/2021/Conference — NeurIPS 2021 Poster_

### Official Review · Reviewer_rhY5 · 2021-07-17

**Rating:** 5
**Confidence:** 5

**Summary:**

This paper studies the adversarial robustness through semi-infinite optimization and non-convex duality theory. By leveraging tools from constrained optimization and duality theory, the authors find the adversarial training formulation is equivalent to learning from a perturbation distribution. The authors propose a constrained learning problem for achieving model robustness while maintaining the nominal performance on clean samples. The authors propose an unconstrained saddle point reformulation for solving the constrained optimization problem via duality, and develop an LMC-based algorithm. Empirically, the authors find the new algorithm can achieve better performance on some standard benchmark tasks.

**Ethical Concerns:**

N.A.

**Limitations And Societal Impact:**

N.A.

**Main Review:**

Pros:
1. The identified connection between constrained formulation in (P-RO) and the statistical counterpart (in Eq. (5)) is interesting, which provides theoretical insights on the constrained problem as well as the trade-off between robustness and accuracy.

Cons:
1. The results in Proposition 3.6 are meaningless. Since $\zeta_{R}(N)$ is the difference between the expected robust risk and the empirical robust risk (defined in Eq. (9a)). $\zeta_{R}(N)$ is relatively large, which suggests $|P^{\star} - \hat{D}^{\star}|$ is large. More specifically, if $\zeta_{R}(N)$ is small, then the model already achieves robust generalization. However, clear robust overfitting has been observed in the literature.
2. The empirical results in Section 6 are not convincing to support the claim 'improved results over state-of-the-art benchmarks' mentioned in the abstract. For example, I would suggest the authors apply tools from RobustBench [85] to evaluate the robust accuracy properly (in order to avoid fake robustness). Also, AWP [46] should be included in section 6.2.

Typo:
1. $(x_n, y_n)$ in Line 268, $x_n$ should be bolded.

Missing reference:
1. Strong duality for non-convex problems, [BLW+2019].

Overall, I think the paper is marginally below the acceptance bar.

[BLW+2019]. Non-Convex Matrix Completion and Related Problems via Strong Duality. Maria-Florina Balcan, Yingyu Liang, David P. Woodruff. Journal of Machine Learning Research, 2019.


**Time Spent Reviewing:**

4

---

> ### Author Response · Authors · 2021-08-10
> **New RobustBench experiments + clarification on generalization bounds**
>
> We thank the reviewer for their thoughtful review. We are glad they find the constrained/statistical duality (eq. (5)) interesting. We hope they will find that our responses below clarify their main concerns.
>
> ## Generalization results
> The reviewer has a point that if the uniform bound on the robust loss $\zeta_R(N)$ defined in (9a) is large, then the generalization guarantees provided in Proposition 3.6 become weaker. However, **this is not a limitation of our method but of the problem itself**: if $\zeta_R(N)$ is large, then learning adversarially robust models requires more samples.
>
> The value of Proposition 3.6 is in establishing that it is possible to learn robust models with good clean accuracy using dual constrained learning at essentially no sample complexity cost. Additionally, it quantifies the idea that the sample complexity of the constrained learning task (PI) depends on how hard it is to learn an adversarially robust model with good clean accuracy. This is embodied by the optimal dual variable bound $\bar{\nu}$.
>
> We emphasize that these are not trivial results. First, (PI) is not a convex problem, so that strong duality does not immediately hold. Hence, establishing that the dual problem is even worth solving is not straightforward. Second, although the dual problem is unconstrained, the guarantees in Proposition 3.6 hold simultaneously for clean and adversarial accuracy, in contrast to regularized approaches. From a learning theoretic perspective, **dual learning therefore does as well as currently used (unconstrained) ERM approaches, but with the added advantage that it provides guarantees on the clean and adversarial accuracy separately rather than on a combination of these objective**.
>
> Finally, note that even if $\zeta_R(N)$ were small, the algorithmic problem of learning adversarially robust models remains due to the difficulty in even evaluating the robust loss $\tilde{l}$ (i.e., in solving the inner maximization problem). We address this issue in Propositions 3.1--3.2.
>
> ## Additional experiments
> Indeed, the main goal of this paper is to build the theoretical foundations of this approach and it therefore focuses on establishing optimality properties and guarantees for this solution. Thus, the experiments are used to showcase the performance expected from the theoretical results. Nevertheless, we agree with the reviewer that accurately reporting robustness results is crucial. As we comment in the paper, it is often the case that models are trained and tested using the same attack, opening up the possibility that they are robust not against adversarial perturbations, but against that specific attack. This is not the case in our experiments, since DALE does not rely on a specific attack during training, but on the optimality results of Propositions 3.1--3.2.
>
> That being said, following the reviewer's suggestion, we are running new experiments using RobustBench. In fact, **using FoolBox (as recommended in the RobustBench documentation) and the AutoAttack adversary on a WideResNet-28-10 yields essentially the same conclusions found on the paper** (compare preliminary results shown below with Table 2): the adversarial accuracy of our method is on par with two strong robust training techniques: TRADES and MART. This supports the fact that our experiments do showcase true effects.
>
> | Algorithm | RobustBench Adv. Acc. (ε=8/255) |
> | :-------: | :---------------: |
> | TRADES | 53.08 |
> | MART | 56.29 |
> | LMC-10 | 54.80 |
>
> We intend to include a complete set of these experiments in the camera-ready paper to make our contribution more concrete and better showcase the effectiveness of training using DALE. We will also include AWP (which we were not previously aware of).
>
> ## Further references
> We also appreciate the reviewer bringing [BLW+2019] to our attention. We will add it to our references.

---

> > ### Comment · Reviewer_rhY5 · 2021-08-18
> > **Quick response**
> >
> > Thank you for your response!
> >
> > 1. **Generalization results**. (1). My main concern is the assumption 3.5 is too strong to hold. As I have mentioned in the earlier review, (9a) + (9b) already implies that if the model could (robustly) fit the training data well then it could generalize (robustly). This is definitely not the case in the adversarial robustness literature (see https://arxiv.org/abs/2002.11569). Given that the assumption is not reasonable, I do not think the dual formulation provides useful insights on this problem; (2). Could you elaborate more on the 'Finally, note that even if $\zeta_{R}(N)$ were small, the algorithmic problem of learning adversarially robust models remains due to the difficulty in even evaluating the robust loss $\tilde{l} $(i.e., in solving the inner maximization problem). We address this issue in Propositions 3.1--3.2.'? E.g., how this evaluation issue is solved? Can the $\mu(\mathbf{x}, y)$ Propositions 3.2 be solved analytically (for practically used network)?
> >
> > 2. **Additional experiments**. Thanks for updating the experiments. Could you confirm the attack method you used for evaluation? I think the standard way to evaluate the model is to apply the '**version= standard**' of autoattack in https://github.com/fra31/auto-attack#autoattack-1.

---

> > > ### Author Response · Authors · 2021-08-20
> > > **More details concerning generalization & experimental results + emphasis on the breadth of our contribution**
> > >
> > > We thank the reviewer for carefully considering our response. We hope that the further details provided below will address your concerns, especially those regarding Assumption 3.5 and the generalization result in Prop. 3.6.
> > >
> > > However, before we discuss these particular results, we want to emphasize that while you have expressed concerns about our generalization results, this result is only one aspect of our contribution.  Indeed, our contributions extend beyond the generalization result, and include:
> > >
> > > - Introducing a new statistical characterization of the adversarial learning problem (Proposition 3.1)
> > > - Deriving an analytical form of the optimal perturbation distribution (Proposition 3.2)
> > > - Proposing a practical algorithm to sample from that distribution as well as learn robust models with good clean performance (Algorithm 1).
> > >
> > > Furthermore, as the reviewers noted, these contributions are both novel in the field of robust learning and -- to quote from your review -- "[provide] theoretical insights on the constrained problem as well as the trade-off between robustness and accuracy."  As such, these results motivate **a myriad of avenues for future research** such as investigating the value of $\gamma$ ($\mu$), improving the sampling procedure to converge faster and accommodate discontinuous distributions, and using the new statistical characterization in Proposition 3.1 to improve our understanding of robust learning and robust overfitting.
> > >
> > > Indeed, we are excited about pursuing each of these avenues in future work and see this paper as a first step towards a complete solution and a better understanding of robust learning. We hope that the details provided in this response (a) answer the reviewer's questions and (b) clarifying the significance of our results in the context of robust learning, a field that has seen thousands of papers written in the last few years and where, hence, novel approaches should be valued.  In the remainder of this response, we seek to address the new points raised in your most recent comment and to highlight how we have revised our paper in line with them. Following these revisions and clarifications, we hope you’ll consider increasing your score.
> > >
> > >
> > > ### Uniform convergence
> > >
> > > With regard to your main concern:
> > >
> > > > "My main concern is the assumption 3.5 is too strong to hold."
> > >
> > > We assume that the reviewer is referring specifically to the statement made in (9a), as there is overwhelming evidence for the uniform convergence property in the non-adversarial case in (9b) (see, e.g., [Bartlett et al., 2017]).  To this end, while we agree in principle that (9a) is relatively strong, there is a growing body of evidence that suggests that **this assumption does in fact hold in practice**, and in particular that it holds for the function class of deep neural networks.  Here are three recent works which show that the sample complexity for the robust learning objective -- as characterized in eq. (9a) of Assumption 3.5 -- can be bounded for practical, commonly-used function classes:
> > >
> > > * [Yin et al., 2019] provides bounds on the Rademacher complexity of the robust learning objective for (binary) linear and nonlinear (e.g., NN) models.  In addition to tightly bounding (with high probability) the sample complexity of robust learning, this work further suggests that the dependence on the dimension of the problem can be overcome, which is critical as common problems in DL deal with high-dimensional data.
> > >
> > > * [Khim and Loh, 2019] show similar bounds on the adversarial Rademacher compleixty for the robust learning using a transformed loss function. Relevantly, these bounds can be specialized to function classes of deep neural networks.
> > >
> > > * [Tu et al., 2019] use a covering-number argument to bound sample complexity for the robust learning objective.  This argument assumes a weak version of Lipschitz continuity for the function class. As it is well-known that neural networks have bounded---and often relatively small---Lipschitz constants (see, e.g., [Fazlyab et al., 2019]), this assumption is reasonable for function classes of deep neural networks.
> > >
> > > Each of these works show that under reasonable assumptions (e.g., on the loss function and the underlying function class), **the sample complexity of robust learning can be bounded with high probability**, which gives ample evidence in support of Assumption 3.5 in our paper.  If the reviewer is interested, we would be happy to provide further references in support of Assumption 3.5 holding in practice.
> > >
> > >
> > > ### Empirical evidence of robust overfitting [Rice et al., 2020]
> > >
> > > With respect to this comment:
> > >
> > > > "This is definitely not the case in the adversarial robustness literature (see https://arxiv.org/abs/2002.11569)."
> > >
> > > While this reference provides **empirical evidence** for robust overfitting, we note that these results are based on **specific, sub-optimal gradient-based attacks** and not on our hybrid sampling optimization procedure.  While it is true that PGD-style attacks find perturbations that improve robust accuracy in practice, such attacks can be prone to overfitting, which leaves open the distinct possibility that **overfitting is due to the implementation of the adversary rather than an inherent drawback of the problem itself**. To illustrate this point, consider the following line of reasoning:
> > >
> > > 1. **Finding worst-case perturbations is hard, even for easy problems.**  First consider that in [Bubeck, et al., 2019], the authors show that show that there exist information-theoretically ''easy'' instances of robust learning problems that are algorithmically hard to solve. Thus even under favorable optimization conditions, it can be challenging to find adversarial perturbations that engender strong robust generalization. Further, it is well-known that the problem of finding worst-case perturbations in the context of DL is non-convex and severely underparameterized, both of which engender a non-favorable optimization landscape [Soltanolkotabi et al., 2018]. Thus, **finding perturbations that result in strong generalization is extremely challenging from an algorithmic point of view in DL.**
> > >
> > > 2. **Robust overfitting may be an algorithmic---rather than an inherent---challenge.**  While the issue of robust overfitting is certainly significant, it is a distinct possibility that this overfitting arises due to the algorithms that are used to solve the problem as well as the non-favorable optimization landscape as described above, rather than being inherent to the robust learning problem itself.  As we show in Appendix A, gradient-based attacks such as PGD can be seen as limiting cases of Algorithm 1. Ultimately, we show that by duality, these attacks can also be thought as extreme approximations of the optimal distribution in Proposition 3.2, where the value of $\mu$ is taken to be so large that the distribution is essentially a point mass on the perturbations that maximize the loss for each data point. These are precisely the points sought out by gradient attacks such as PGD. In other words, these gradient-based attacks use a **critically undersmoothed approximation of the optimal distribution** $\lambda^\star$, thus incurring the risk of training using perturbations stronger than necessary, degrading the generalization properties.
> > >
> > > 3. **Our theory specifies how previous attacks are sub-optimal, suggesting why they overfit.** Motivated by the difficulty of sampling high-dimensional discontinuous distributions, the algorithm we develop and the corresponding numerical results use what is likely an **oversmoothed approximation of the optimal distribution** (taking $\mu \equiv 0$). This provides a principled form of regularization against the issue of robust overfitting, which from the experiments in the reference mentioned by the reviewer [Rice et al., 2020], appears to be necessary. Based on our results, it would seem that oversmoothing can be more beneficial in this setting than undersmoothing, which provides additional reasons for using $\mu \equiv 0$.
> > >
> > > To summarize, based solely on the empirical results in [Rice et al., 2020], it is hard to unilaterally conclude that adversarially robust generalization is impossible, especially given that this is at odds with the theoretical results discussed in the previous section. Indeed, our theoretical results indicate that the adversaries used in [Rice et al., 2020] are suboptimal in ways that leave them susceptible to overfitting and opens the distinct possibility that Assumption 3.5 is not at odds with the work in [Rice et al., 2020]. We agree with the reviewer and apologize that this point was not made explicit in the main text. We have already incorporated a new section in our manuscript that explicitly addresses this point, in line with the comments made by the reviewer.

---

> > > > ### Author Response · Authors · 2021-08-20
> > > > **More details concerning the dual formulation & our experiments**
> > > >
> > > > ### Dual formulation
> > > >
> > > > With respect to this comment:
> > > >
> > > > > "Given that the assumption is not reasonable, I do not think the dual formulation provides useful insights on this problem."
> > > >
> > > > We are not sure exactly what the reviewer means here.  Below, we reply to two possible interpretations of this comment; please do not hesitate to clarify if we have not addressed your point.
> > > >
> > > > * If the reviewer is referring to the duality result in Proposition 3.1., we emphasize that this result has no bearing on generalization result presented later in the paper since it is not concerned with empirical approximations.
> > > >
> > > > * On the other hand, if the reviewer is referring to the dual learning result in Proposition 3.6., we reiterate that the main contribution of this result is establishing that it is possible to learn models with strong clean and robust performance. This result is not straightforward and we feel that it is somewhat counterintuitive, since there is strong evidence across numerous papers which claim that clean and robust performance are at odds (see, e.g. [Tsipras et al., 2019; Zhang et al., 2019]). Indeed, numerous adversarial learning procedures were developed to mitigate this issue (see the references [59--62] in our paper), although the generalization properties of their solutions have not been theoretically established.
> > > >
> > > >   Additionally, note that Proposition 3.6 shows that there are factors beyond uniform convergence that affect our ability to trade-off clean accuracy for robustness. In particular, this proposition establishes the highly non-trivial fact that the clean accuracy is unaffected by the robust learning objective, even though the constrained learning problem couples these two quantities. This coupling appears in terms of the sensitivity of the objective to the constraint, i.e., on the difficulty of the trade-off. Hence, even if we could learn robustly and classically separately, it could still be hard to learn them together.
> > > >
> > > >   Moreover, we want to emphasize again that, regardless of Assumption 3.5, these are not trivial results: (PI) is a non-convex problem, so that strong duality is not immediate and the guarantees hold simultaneously for clean and adversarial accuracy, in contrast to regularized approaches.
> > > >
> > > > ### Evaluating the robust loss
> > > >
> > > > With respect to this comment:
> > > >
> > > > > "Could you elaborate more on the 'Finally, note that even if $\zeta_R(N)$ were small, the algorithmic problem of learning adversarially robust models remains due to the difficulty in even evaluating the robust loss $\tilde{l}$
> > > > (i.e., in solving the inner maximization problem). We address this issue in Propositions 3.1--3.2.'? E.g., how this evaluation issue is solved? Can the $\mu(x,y)$ Propositions 3.2 be solved analytically (for practically used network)?."
> > > >
> > > > Note that the value of $\mu$ is fully determined by $\gamma$, which is directly related to the the smoothness of the optimal distribution (it is, in fact, equivalent to a bound on its L2-norm). While it may be possible to determine an analytic form for $\mu$ in certain cases, the reviewer has a point that it is certainly not trivial to do so (particularly for neural networks).  And even if it were, the resulting distribution for any $\mu \neq 0$ is discontinuous and sampling from such distributions in high-dimensional settings is a challenge in and of itself. It is, in fact, an area of active research in Monte Carlo methods (see [Nishimura et al., 2020] and references therein).
> > > >
> > > > That being said, the fact that there exists a $\mu$ (or, more precisely, a $\gamma$) for which we can obtain a **closed-form, analytical representation of the optimal perturbation distribution $\lambda^\star$ is certainly not trivial**. In fact, it is only true under conditions (see Proposition hypotheses) that, though mild, are not general. Hence, we believe that establishing this result is already a contribution, though we agree that further characterizing this distribution is an important avenue for future research.
> > > >
> > > >
> > > > ## Additional experiments details
> > > >
> > > > With respect to your comment:
> > > >
> > > > > "Could you confirm the attack method you used for evaluation? I think the standard way to evaluate the model is to apply the 'version= standard' of autoattack"
> > > >
> > > > Yes, this is exactly what we did.  We will clarify this point in the camera-ready version.
> > > >
> > > > ---
> > > >
> > > > ## References
> > > >
> > > > [Bartlett et al., 2017] Peter L. Bartlett, Dylan J. Foster, and Matus J. Telgarsky. "Spectrally-normalized margin bounds for neural networks." In Advances in Neural Information Processing Systems 30 (Neurips 2017)
> > > >
> > > > [Yin et al., 2019] Dong Yin, Ramchandran Kannan, and Peter Bartlett. "Rademacher complexity for adversarially robust generalization." In International Conference on Machine Learning, pages 7085–7094. PMLR, 2019.
> > > >
> > > > [Khim and Loh, 2019] Justin Khim and Po-Ling Loh. "Adversarial risk bounds via function transformation." arXiv preprint arXiv:1810.09519 (2018).
> > > >
> > > > [Tu et al., 2019] Zhuozhuo Tu, Jingwei Zhang, and Dacheng Tao. "Theoretical analysis of adversarial learning: A minimax approach."  In the 33rd Conference on Neural Information Processing Systems (NeurIPS 2019).
> > > >
> > > > [Fazlyab et al., 2019] Mahyar Fazlyab, Alexander Robey, Hamed Hassani, Manfred Morari, George J. Pappas. "Efficient and accurate estimation of lipschitz constants for deep neural networks." In the 33rd Conference on Neural Information Processing Systems (NeurIPS 2019).
> > > >
> > > > [Montasser et al., 2019] Omar Montasser, Steve Hanneke, and Nathan Srebro. "Vc classes are adversarially robustly learnable, but only improperly." Conference on Learning Theory. PMLR, 2019.
> > > >
> > > > [Cullina et al., 2018] Daniel Cullina, Arjun Nitin Bhagoji, and Prateek Mittal. "PAC-learning in the presence of evasion adversaries." 32nd Conference on Neural Information Processing Systems (NeurIPS 2018).
> > > >
> > > > [Rice et al., 2020] Leslie Rice, Eric Wong, and Zico Kolter. "Overfitting in adversarially robust deep learning." International Conference on Machine Learning. PMLR, 2020.
> > > >
> > > > [Soltanolkotabi et al., 2018] Mahdi Soltanolkotabi, Adel Javanmard, and Jason D Lee. Theoretical insights into the optimization landscape of over-parameterized shallow neural networks. IEEE Transactions on Information Theory, 65(2):742–769, 2018.
> > > >
> > > > [Bubeck, et al., 2019] Sébastien Bubeck, Yin Tat Lee, Eric Price, and Ilya Razenshteyn. "Adversarial examples from computational constraints." International Conference on Machine Learning. PMLR, 2019.
> > > >
> > > > [Tsipras et al., 2019] Dimitris Tsipras, Shibani Santurkar, Logan Engstrom, Alexander Turner, Aleksander Madry. "Robustness may be at odds with accuracy." In International Conference on Learning Representations (ICLR 2019)
> > > >
> > > > [Zhang et al., 2019] Hongyang Zhang, Yaodong Yu, Jiantao Jiao, Eric Xing, Laurent El Ghaoui, Michael Jordan. "Theoretically principled trade-off between robustness and accuracy." International Conference on Machine Learning. PMLR, 2019.
> > > >
> > > > [Nishimura et al., 2020] Akihiko Nishimura, David B Dunson, and Jianfeng Lu. "Discontinuous hamiltonian monte carlo for discrete parameters and discontinuous likelihoods." Biometrika, 107(2):365–380, 2020.

---

> > > > > ### Comment · Reviewer_rhY5 · 2021-08-23
> > > > > **Quick response**
> > > > >
> > > > > Thanks for the detailed response.
> > > > >
> > > > > >Evaluating the robust loss
> > > > >
> > > > > I was referring to whether the approaches proposed in the paper could resolve the problem 'the algorithmic problem of learning adversarially robust models remains due to the difficulty in even evaluating the robust loss' mentioned in earlier author response. Based on the updated response, it seems that this issue cannot be solved in this submission. Please correct me if I am wrong. If so, I would suggest the authors avoid overclaiming on this point.

---

> > > > > > ### Author Response · Authors · 2021-08-25
> > > > > > **We did not claim to algorithmically solve the problem of evaluating the robust loss + by construction, uniform convergence cannot be tested on standard benchmarks**
> > > > > >
> > > > > > $\DeclareMathOperator*{\argmax}{argmax}$
> > > > > >
> > > > > > We thank the reviewer for this engaging discussion.
> > > > > >
> > > > > > **Regarding our contribution.**  We believe that there has been a misunderstanding about the contribution of our work.  We **never** claimed to be able to algorithmically resolve the problem of **evaluating** the robust loss, i.e. to solve the optimization problem:
> > > > > >
> > > > > > $$ \delta^\star \in\argmax_{\delta\in\Delta(\epsilon)} \ell(f_\theta(x+\delta), y) \quad\text{where } \Delta = \{\delta\in\mathbb{R}^d : ||\delta||_p \leq \epsilon\}.$$
> > > > > >
> > > > > > On the contrary, our contributions (as per lines 60-71 of the paper) include (1) lifting the well-known minimax robustness problem to a stochastic optimization problem over a specific, non-atomic perturbation distribution, (2) proposing an algorithm to (empirically) tackle this problem, (3) showing that our algorithm performs on-a-par with SOTA algorithms, and (4) providing PAC-style generalization results for our algorithm.
> > > > > >
> > > > > > Indeed, the problem of evaluating the robust loss is highly non-convex for common function classes (e.g. deep neural networks), and thus **extremely challening to solve in practice.**  Thus, rather than attempting to solving this nonconvex problem, the main idea in our work is to transform the minimax problem
> > > > > >
> > > > > > $$ \min_\theta \mathbb{E}\left[ \max_{\delta\in\Delta(\epsilon)} \ell(f_\theta(x+\delta), y) \right] $$
> > > > > >
> > > > > > into a distributional problem over perturbation distributions.  This lifts the nonconvex problem shown above to a functional optimization problem over probability distributions.  Further, we show the nontrivial fact that strong duality holds for these problems.  Thus, our approach is to completely recharacterize the robust learning problem, which we feel will open up numerous avenues for future study and will be of interest to theorists and practioners due to the prevalence of robust learning problems (see Appendix A for details).  If the reviewer feels that these contributions were not clearly enumerated, we would be happy to revise the relevant parts of our paper, and would appreciate any additional feedback concerning how we can most effectively do this.
> > > > > >
> > > > > > **Regarding the PAC-style results.**  We also want to emphasize that the generalization results and the uniform convergence properties enumerated in our Assumption 3.5 and Prop. 3.6 are theoretical constructions upon which much of standard PAC learning theory is based.  For reference, such topics are covered in depth in Chapters 4-6 of the following reference:
> > > > > >
> > > > > > > Shai Shalev-Shwartz and Shai Ben-David. Understanding machine learning: From theory to algorithms. Cambridge university press, 2014.
> > > > > >
> > > > > > These kinds of PAC generalization results are meant to provide worst-case bounds for any possible data distribution and for any hypothesis in the hypothesis class, so that under very general conditions, it can be shown that (with high probability) a given algorithm achieves good performance.  For this reason, computationally evaluating whether uniform convergence hold for a particular benchmark (e.g. CIFAR-10) is infeasible, as it would involve iterating over all functions in an infinite functional space.  For this reason, it would not be possible to evaluate the uniform convergence property even in the non-adversarial setting.  Indeed we emphasize that such a evaluation misses the point of these kinds of results.  These results are intended to provide theoretical evidence of (1) the computational hardness of the problem and (2) our approach's ability to overcome this hardness.  To this end, we would like to point the reviewer toward three recent papers published in NeurIPS which also consider PAC-style gaurantees in the adversarial setting:
> > > > > >
> > > > > > > Daniel Cullina, Arjun Nitin Bhagoji, and Prateek Mittal. "PAC-learning in the presence of evasion adversaries."  32nd Conference on Neural Information Processing Systems (NeurIPS 2018), Montreal, Canada.
> > > > > >
> > > > > > > Luiz Chamon and Alejandro Ribeiro. "Probably approximately correct constrained learning." 34th Conference on Neural Information Processing Systems (NeurIPS 2020), Vancouver, Canada.
> > > > > >
> > > > > > > Omar Montasser, Steve Hanneke, and Nathan Srebro. "Reducing Adversarially Robust Learning to Non-Robust PAC Learning."  34th Conference on Neural Information Processing Systems (NeurIPS 2020), Vancouver, Canada.
> > > > > >
> > > > > > In a similar vein to our paper, each of these works studies the computational hardness of adversarial robustness and robust algorithms from a learning theoretic point of view.  In general, as evinced by this line of work in the above papers and the attention received by robust learning problems in recent years, it is clear that there is a community-level interest in understanding the theoretical limits of robust learning through PAC-learning type guarantees.  Our approach directly follows this line of work, as we use learning theoretic arguments to recharacterize the adversarial robustness problem and to study its fundamental limits.

---

> > > > > > > ### Author Response · Authors · 2021-08-28
> > > > > > > **Touching base + have we addressed your concerns?**
> > > > > > >
> > > > > > > We wanted to touch base again following this fruitful discussion.  We hope that throughout the course of this discussion, we have addressed your main concerns.  To summarize so far:
> > > > > > >
> > > > > > > * We seem to agree that while adversarially robust generalization is possible, the assumptions that underly much of the learning theory literature (i.e. the uniform convergence assumption on the clean and robust risks) may not hold on practical datasets like CIFAR and MNIST.  We emphasize that while strong assumptions are necessary to prove the general, distribution-independent learning theoretic bounds such as Prop. 3.5, these assumptions are not needed for any other part -- whether theoretical or experimental -- of our paper.
> > > > > > >
> > > > > > > * Our contribution is **not** to solve the nonconvex problem of evaluating the robust loss.  As noted by the other reviewers, our contributions include "presenting a novel take on adversarial robustness" (Reviewer JwwU) in which we prove "a novel connection between the well-known adversarial training problem and a statistical problem over perturbation distributions" (Reviewer v1iK).  The derivation of our algorithm is guided by this new presepective, and we show that our method is competitve with state-of-the-art results.
> > > > > > >
> > > > > > > * We **added experiments** on RobustBench using AutoAttack following your suggestions, which we feel has strengthened our empirical results.
> > > > > > >
> > > > > > > If there is anything else that we can clarify, **we would be more than happy to continue this discussion.**

---

> > > > > > > > ### Comment · Reviewer_rhY5 · 2021-08-28
> > > > > > > > **Quick response**
> > > > > > > >
> > > > > > > > Thanks for the response. I have several clarification questions:
> > > > > > > >
> > > > > > > > What's the benefit of '**lifting the well-known minimax robustness problem to a stochastic optimization problem over a specific, non-atomic perturbation distribution**'? For example, compared with the original minimax robustness problem, does the algorithm (Algorithm 1) provide convergence guarantees for solving this new problem? (could consider even the linear case.)
> > > > > > > >
> > > > > > > > From my understanding, for solving the new problem, the authors considered a simplified setting ($\mu_{n} = 0$ in Line 275). Is there any quantification of this mismatch? Also, is the LMC proved to solve the subproblem (solve $\delta_i$)?

---

> > > > > > > > > ### Author Response · Authors · 2021-08-30
> > > > > > > > > **Benefits of lifting the minimax problem + further details concerning Langevin Monte Carlo methods**
> > > > > > > > >
> > > > > > > > > $\DeclareMathOperator*{\argmax}{argmax}$
> > > > > > > > >
> > > > > > > > > Thanks for your questions and for your continued engagement with our paper.
> > > > > > > > >
> > > > > > > > > > "What's the benefit of 'lifting the well-known minimax robustness problem to a stochastic optimization problem over a specific, non-atomic perturbation distribution'?"
> > > > > > > > >
> > > > > > > > > There are several practical benefits of the minimax robustness formulation from Prop. 3.1.  Indeed, we emphasize that the equivalence of these two problems is a nontrivial mathematical fact, which we believe will be of independent interest to many in the adversarial robustness community.  Aside from this, we list other practical benefits below:
> > > > > > > > >
> > > > > > > > > * The most important consequence of lifting the standard minimax problem
> > > > > > > > >
> > > > > > > > >     $$ \min_\theta \mathbb{E}\left[ \max_{\delta\in\Delta(\epsilon)} \ell(f_\theta(x+\delta), y) \right] $$
> > > > > > > > >
> > > > > > > > >     is that it **transforms the inner problem from a nonconcave optimization problem to a linear optimization problem** over probability distributions (see lines 189-191).  To see this, observe that the inner problem
> > > > > > > > >
> > > > > > > > >     $$ \delta^\star\in \argmax_{\delta\in\Delta(\epsilon)} \ell(f_\theta(x+\delta), y) $$
> > > > > > > > >
> > > > > > > > >     is **nonconcave** when $f_\theta$ parameterizes a deep neural network (or any nonlinear model for that matter, e.g., kernel models).  Thus, as we dicussed in a previous comment titled "More details concerning generalization & experimental results," solving the inner maximization problem to a global optimimum $\delta^\star$ is extremely challenging; indeed, the best that one can hope for with gradient-based algorithms is a stationary point, which may be far from the true maximizer.  On the other hand, by lifting the minimax problem, we transform this nonconcave problem into a **linear optimization problem** over probability distributions:
> > > > > > > > >
> > > > > > > > >     $$ \min_\theta \max_{\lambda\in\mathcal{P}^q} \mathbb{E} \left[ \mathbb{E}_{\delta\sim \lambda(\delta|x,y)} \ell(f_\theta(x,y)) \right]. $$
> > > > > > > > >
> > > > > > > > >     Given this novel insight, the inner problem:
> > > > > > > > >
> > > > > > > > >     $$ \lambda^\star\in \argmax_{\lambda\in\mathcal{P}^q} \mathbb{E} \left[ \mathbb{E}_{\delta\sim \lambda(\delta|x,y)} \ell(f_\theta(x,y)) \right] $$
> > > > > > > > >
> > > > > > > > >      can be seen as a problem of sampling from the "worst-case" distribution $\lambda^\star$, which can be addressed by statistical sampling based techniques such as the Langevin Monte Carlo technique we use in Algorithm 1.
> > > > > > > > >
> > > > > > > > > * The reformulation in Prop. 3.1 also allows us to characterize the optimal perturbation distribution required to learn robustly.  That is, in some special cases, we derive **a closed-form expression for the distribution that maximizes the objective of the inner problem** in the lifted problem.  In particular, we show in Prop. 3.2 that when the map $(x,y) \mapsto \ell(f_\theta(x),y)$ is in $L^2$, eq. (6) gives this closed-form expression for the "lifted" minimax problem.  This is a significant result that -- as far as we know -- cannot be replicated for the original minimax problem.
> > > > > > > > >
> > > > > > > > > * Finally, an important consequence of the reformulation in Prop. 3.1 is a **precise, guaranteed control over the approximation of the robust learning problem**.  Note that gradient-based methods for solving the original minimax problem may obtain stationary points, which can be arbitrarily far from the global optimum. Moreover, most theoretical results on neural network training rely on their severe overparametrization, which is not the case for the inner maximization problem in the original minimax problem (as discussed in detail in our previous comment "More details concerning generalization & experimental results"), and thus the weak guarantees that exist for convergence of gradient-based methods for training overparameterized neural networks do not apply when solving the original inner maximization problem to find worst-case perturbations.
> > > > > > > > >
> > > > > > > > >     On the other hand, in our approach, there is an explicit parameter, namely $\mu$, that controls the quality of our approximation to the global optimum we obtain for the inner problem. This is due to the fact that **(1)** we have found a stochastic optimization analog of the deterministic robust learning problem (Prop. 3.1); **(2)** we have exactly characterized (under mild conditions) the optimal perturbation distribution $\lambda^\star$; and **(3)** it is possible to sample from that distribution using techniques that, though similar to optimization methods, behave quite differently.  In this way, we can control the quality of our solution by varying this parameter $\mu$; **no such guarantees or control is available for the original minimax problem.**
> > > > > > > > >
> > > > > > > > > > "From my understanding, for solving the new problem, the authors considered a simplified setting ($\mu_n=0$ in Line 275). Is there any quantification of this mismatch? Also, is the LMC proved to solve the subproblem (solve $\delta_i$)?"
> > > > > > > > >
> > > > > > > > > Let us start by clarifying that, though the choice of $\mu=0$ does simplify the sampling procedure, it is not a **simplification of the solution; it is simpliciation of our approximation of the solution**. To this end, Prop. 3.2 shows that there exists a parameter $\mu$ for which the solution is exact.  We note that such a parameter is not known *a priori*; while we feel that investigating the theoretical properties of this approximation is beyond the scope of this work, our numerical results provide evidence that it is quite effective in practice, by achieving (and sometimes improving over) state-of-the-art results.  Reaching a more complete understanding of how to obtain or learn $\mu$ is a direction that we plan to pursue in future work.
> > > > > > > > >
> > > > > > > > > With respect to LMC, let us be clear that **our approach does not obtain (or even seek to obtain) the worst-case perturbation $\delta_i^\star$ that would achieve the maximum in the original robust learning problem**. As we argued in our paper and in the preceeding discussion, this is likely to be hard in general (see "More details concerning generalization & experimental results" in our previous comment). Prop. 3.1 allows us to circumvent the need for finding worst-case perturbations by sampling from the optimal distribution $\lambda^\star$. Notice, however, that $\lambda^\star$ is the distribution that leads the optimization problem to the solution of the robust learning problem and need not be concentrated on any specific worst-case point.
> > > > > > > > >
> > > > > > > > > As for LMC dynamics, its convergence properties have been widely studied in the literature and we point to several papers that examine its ability to sample from specific distributions. For illustration here, we mention the work in the following reference from NIPS 2015:
> > > > > > > > >
> > > > > > > > > > Sebastien Bubeck, Ronen Eldan, and Joseph Lehec. "Finite-time analysis of projected Langevin Monte Carlo." Advances in Neural Information Processing Systems. 2015.
> > > > > > > > >
> > > > > > > > > in which the LMC is shown to obtain samples from $\lambda^\star$ in a polynomial number of steps $N = \tilde{\Omega}(\text{poly}(n/\epsilon^2))$, where $n$ is the data dimension and $\epsilon$ is the TV distance to its stationary distribution (in our case $\lambda^\star$). Other works have studied LMC (or more generally, Hamiltonian Monte Carlo) methods under different assumptions and conditions. Note that convergence guarantees for sampling are often considerably stronger than those obtained for optimization algorithms such as SGD; if the reviewer is interested, the following reference is a good starting point for such results:
> > > > > > > > >
> > > > > > > > > > Maxim Raginsky, Alexander Rakhlin, Matus Telgarsky. Non-Convex Learning via Stochastic Gradient Langevin Dynamics: A Nonasymptotic Analysis. COLT, 2017.
> > > > > > > > >
> > > > > > > > > We note that there are several alternative algorithms that may improve convergence and performance, such as
> > > > > > > > >
> > > > > > > > > > László Lovász. "Hit-and-run mixes fast." Mathematical programming 86.3 (1999): 443-461.
> > > > > > > > >
> > > > > > > > > > Akihiko Nishimura, David B. Dunson, and Jianfeng Lu. "Discontinuous Hamiltonian Monte Carlo for discrete parameters and discontinuous likelihoods." Biometrika 107.2 (2020): 365-380.
> > > > > > > > >
> > > > > > > > > We defer the exploration of these more advanced methods to future work seen as (1) LMC already yields very good performance in our numerical experiments and (2) LMC allows us to draw connections to existing robust learning procedures (see Appendix A in supplementary materials).

---

> > > > > > > > > > ### Comment · Reviewer_rhY5 · 2021-08-31
> > > > > > > > > > **response**
> > > > > > > > > >
> > > > > > > > > > Thanks for the response.
> > > > > > > > > >
> > > > > > > > > > First of all, [1] considers the convex setting, and the results cannot provide justifications on the LMC considered in this paper. Also, [2] needs to make the dissipative assumption, and I do not think the problem considered in this paper satisfies this assumption.
> > > > > > > > > >
> > > > > > > > > > From my understanding, there is no theoretical guarantee for solving the subproblem in the lifted formulation. Please correct me if I am wrong. Thanks.
> > > > > > > > > >
> > > > > > > > > >
> > > > > > > > > >
> > > > > > > > > > [1]. Sebastien Bubeck, Ronen Eldan, and Joseph Lehec. "Finite-time analysis of projected Langevin Monte Carlo." Advances in Neural Information Processing Systems. 2015.
> > > > > > > > > > [2]. Maxim Raginsky, Alexander Rakhlin, Matus Telgarsky. Non-Convex Learning via Stochastic Gradient Langevin Dynamics: A Nonasymptotic Analysis. COLT, 2017.

---

> > > > > > > > > > > ### Author Response · Authors · 2021-08-31
> > > > > > > > > > > **Further details concerning convergence of LMC + this is getting well beyond the scope of our paper**
> > > > > > > > > > >
> > > > > > > > > > > Thanks for your response.
> > > > > > > > > > >
> > > > > > > > > > > We are not sure what the reviewer means by "the convex setting."  We give answers with respect to two different interpretations below:
> > > > > > > > > > >
> > > > > > > > > > > * If the convexity is in reference to the lifted inner maximization problem described in our previous response titled "Benefits of lifting the minimax problem," we encourage the reviewer to reread this response, in which we explained that by lifting the minimax problem, we transform the non-cocave inner problem $\delta^\star\in \argmax_{\delta\in\Delta(\epsilon)} \ell(f_\theta(x+\delta), y)$ into a linear program $\lambda^\star\in \argmax_{\lambda\in\mathcal{P}^q} \mathbb{E} \left[ \mathbb{E}_{\delta\sim \lambda(\delta|x,y)} \ell(f_\theta(x,y)) \right]$.  Note that **linear programs are concave programs**; for reference, see Section 4.3 in
> > > > > > > > > > >
> > > > > > > > > > >     > Stephen P. Boyd and Lieven Vandenberghe. Convex optimization. Cambridge university press, 2004.
> > > > > > > > > > >
> > > > > > > > > > >     Thus, while indeed the lifted composite problem is generally nonconvex due to the fact that the parameter space $\Theta$ may be nonconvex (as is the case for the class of neural nets), the lifted inner problem remains a concave maximization problem.
> > > > > > > > > > >
> > > > > > > > > > > * If, on the other hand, by "the convex setting" the reviewer means that the function $f$ in [1] is assumed to be convex, which in turn induces a log-concave density $(1/Z) \exp(-f(x))\mathbb{1}(x\in \mathcal{K})$, we agree that there are cases for which the convergence results may not apply when $\Theta$ is nonconvex.  Note that for linear models $f_\theta(x) = \theta^\top x$, this assumption clearly holds, and thus, to answer your question from the post titled "Quick response" from Aug. 28, convergence of LMC is **guaranteed** in this special case.
> > > > > > > > > > >
> > > > > > > > > > >     To this end, we emphasize that **the point of our paper is not to prove convergence guarantees for nonconvex/nonconcave problems**.   In our paper, we provide a short overview of well-known LMC results in straightforward cases that is in no way meant to be exhaustive or complete. As we wrote in our previous response
> > > > > > > > > > >
> > > > > > > > > > >     > [...] we point to several papers that examine its ability to sample from **specific distributions**. For illustration here, we mention the work in [Sebastien Bubeck, ...]
> > > > > > > > > > >
> > > > > > > > > > >     That is because **our method is in no way dependent on a particular sampling technique** and we in fact remark in the text that other, more advanced samplers may provide even better performance and/or guarantees (lines 276--278). As we have remarked before, we use LMC mainly because it **(1) yields very good performance in numerical experiments** and **(2) allows us to draw connections to existing robust learning procedures** (see Appendix A in supplementary materials). For that reason, we do not delve into the details of the properties of MCMC methods in this paper, though we believe it is an interesting avenue of future work. That being said, the reviewer is correct that sampling is often studied in the log-concave setting and that it has not been explicitly investigated in the specific context of robust learning.  Indeed, our work on robust learning provides a new compelling reason to study this setting beyond the log-concave setting. We believe the new perspective presented in this paper will bring forward interest in this intersection.
> > > > > > > > > > >
> > > > > > > > > > >     Nevertheless, **recent works have shown evidence of the advantages of LMC in non-convex settings**. We point to
> > > > > > > > > > >
> > > > > > > > > > >     > [Maxim Raginsky, Alexander Rakhlin, Matus Telgarsky. Non-Convex Learning via Stochastic Gradient Langevin Dynamics: A Nonasymptotic Analysis. COLT, 2017]
> > > > > > > > > > >
> > > > > > > > > > >     as an example, but the following work
> > > > > > > > > > >
> > > > > > > > > > >     > [Yi-An Ma, Yuansi Chen, Chi Jin, Nicolas Flammarion, Michael I. Jordan. Sampling can be faster than optimization, PNAS, 2019]
> > > > > > > > > > >
> > > > > > > > > > >     may be a better reference in the sampling context. **Convergence properties of LMC in non-log-concave settings** are also studied, e.g., in
> > > > > > > > > > >
> > > > > > > > > > >     > Difan Zou, Pan Xu, Quanquan Gu. Sampling from Non-Log-Concave Distributions via Stochastic Variance-Reduced Gradient Langevin Dynamics. AISTATS, 2019
> > > > > > > > > > >
> > > > > > > > > > >     > Oren Mangoubi, Nisheeth K. Vishnoi. Nonconvex sampling with the Metropolis-adjusted Langevin algorithm. COLT, 2019
> > > > > > > > > > >
> > > > > > > > > > >     > Rong Ge, Holden Lee, Andrej Risteski. Simulated Tempering Langevin Monte Carlo II: An Improved Proof using Soft Markov Chain Decomposition
> > > > > > > > > > >
> > > > > > > > > > >     These references notwithstanding, the convergence properties of **LMC dynamics for non-log-concave distributions remains an area of active research, well-beyond the scope of this paper**.
> > > > > > > > > > >
> > > > > > > > > > > To summarize, while the question of the convergence of sampling algorithms like LMC is interesting, it is far beyond the scope of our paper.  We hope that this discussion has cleared up any remaining confusion, and given this, we hope that the reviewer would consider raising their score.

---

> > > > > > > > > > > > ### Comment · Reviewer_rhY5 · 2021-09-01
> > > > > > > > > > > > **quick response**
> > > > > > > > > > > >
> > > > > > > > > > > > Thanks for the response.
> > > > > > > > > > > >
> > > > > > > > > > > > > About the convex setting.
> > > > > > > > > > > >
> > > > > > > > > > > > The 'convex setting' I was referring to is 'We show that LMC allows to sample in polynomial time from a posterior distribution restricted to a convex body and with concave log-likelihood.' as mentioned in '[1]. Sebastien Bubeck, Ronen Eldan, and Joseph Lehec. "Finite-time analysis of projected Langevin Monte Carlo." Advances in Neural Information Processing Systems. 2015.'.
> > > > > > > > > > > >
> > > > > > > > > > > > > About the guarantee of LMC.
> > > > > > > > > > > >
> > > > > > > > > > > > As mentioned in my earlier reply, 'From my understanding, there is no theoretical guarantee for solving the subproblem in the lifted formulation. Please correct me if I am wrong.', could the authors directly answer this question? I know there are results for non-convex settings, but I do not think the subproblem considered in this paper satisfies the assumptions of those settings.
> > > > > > > > > > > >
> > > > > > > > > > > > Again, to highlight my previous comment: there is no theoretical guarantee for solving the subproblem in the lifted formulation. Therefore, in terms of solving the real-world problem, the proposed algorithm in this paper does not have theoretical guarantees.

---

> > > > > > > > > > > > > ### Author Response · Authors · 2021-09-02
> > > > > > > > > > > > > **Convergence guarantees are outside of the scope of our paper**
> > > > > > > > > > > > >
> > > > > > > > > > > > > ## Regarding our contribution & convergence analysis
> > > > > > > > > > > > >
> > > > > > > > > > > > > In this response, we will do our best to answer your questions.  However, before we do that, we want to emphasize that
> > > > > > > > > > > > >
> > > > > > > > > > > > > > **Analyzing the convergence of Algorithm 1 is beyond the scope of our paper.**
> > > > > > > > > > > > >
> > > > > > > > > > > > > Here's why:
> > > > > > > > > > > > >
> > > > > > > > > > > > > * **This paper is not about convergence guarantees.** Our paper is about studying the robust learning optimization problem; it is *not* about analyzing the convergence of a particular algorithm.  All of our theoretical results apply to the robust learning problem itself, and are not specific to Algorithm 1 or to any other algorithm.  Indeed, as we remark in lines 276-279, other algorithms could be used to solve the dual problem (P-DALE).  We chose Langevin Monte Carlo for two reasons: (1) it yields very good performance in numerical experiments and (2) it allows us to draw connections to existing robust learning procedures.
> > > > > > > > > > > > >
> > > > > > > > > > > > > * **We never claimed to provide convergence guarantees.**  At no point in our paper or response did we claim to provide convergence guarantees.  We emphasize that providing such guarantees is the subject of active and ongoing research, wherein there are many open problems.  With this in mind, we believe that our paper should be judged based on how we validate the claims in our paper; it should not be judged based on claims that are outside of the scope of our contribution.
> > > > > > > > > > > > >
> > > > > > > > > > > > > * **Algorithm 1 is a new algorithm.**  None of the papers we have cited in our responses provide convergence guarantees for Algorithm 1 in full generality; as we've evidently just proposed this algorithm in this paper, we emphasize: **This is an open problem.**  Algorithm 1 is a new algorithm to which existing analyses do not fully apply.  Proving the convergence of Algorithm 1 in full generality for nonconvex function classes would be an immense contribution, worthy of several further papers.
> > > > > > > > > > > > >
> > > > > > > > > > > > > Given these points, we again emphasize that an analysis of the convergence of Algorithm 1 is well beyond the scope of our contribution.
> > > > > > > > > > > > >
> > > > > > > > > > > > > ## Regarding "real-world problems"
> > > > > > > > > > > > >
> > > > > > > > > > > > > To this end, regarding your comment:
> > > > > > > > > > > > >
> > > > > > > > > > > > > > "Therefore, in terms of solving the real-world problem, the proposed algorithm in this paper does not have theoretical guarantees."
> > > > > > > > > > > > >
> > > > > > > > > > > > > The reviewer seems to be claiming that if we do not prove that Algorithm 1 converges to the global optimum, then it is of no use in "real-world problems" due to an apparent lack of theoretical guarantees.  We strongly disagree with this statement.  Here's why:
> > > > > > > > > > > > >
> > > > > > > > > > > > > If the reviewer stands by this claim, then we emphasize that **no adversarial training algorithm possesses such guarantees.**  Consider that PGD applied in the setting of adversarial perturbations is only known to converge in **highly simplified settings.**  In particular, the only convergence result that we know of for PGD in the nonconvex setting is the work of [Gao et al., 2019], which makes several highly-restrictive assumptions, including that data is uniformly distributed on the unit sphere.  Clearly this setting cannot characterize the span of "real-world problems."
> > > > > > > > > > > > >
> > > > > > > > > > > > > All of this being said, we again emphasize that convergence guarantees for this problem are well beyond the scope of this paper, and that the lack of such guarantees neither lessens our contribution nor does it diminish the potential of our work toward addressing "real-world problems."
> > > > > > > > > > > > >
> > > > > > > > > > > > > ---
> > > > > > > > > > > > >
> > > > > > > > > > > > > ## Speculation concerning the convergence of Algorithm 1
> > > > > > > > > > > > >
> > > > > > > > > > > > > Throughout our responses, we have provided a fairly extensive review of the literature concerning the convergence of sampling-based methods.  The goal of this has been to address the questions raised in your post-rebuttal comments.  Ultimately, over the course of these responses, we have provided intuition for our algorithm and speculated on possible avenues toward proving general convergence results.  While interesting, these comments will not be incorporated as part of our paper, as they are well outside the scope of this work.  However, due to the interest that the reviewer has shown on this topic, in this post we will attempt to compile some thoughts concerning the convergence of Algorithm 1.
> > > > > > > > > > > > >
> > > > > > > > > > > > > * When $\Theta$ is a compact space of **linear** classifiers, so that $f_\theta(x) = \theta^\top x$, the convergence results of [Bubeck et al., 2015] apply, and the LMC procedure will produce samples from $\lambda^\star$ after sufficiently many iterations.  Furthermore, in this case, the outer minimization problem is linear, and therefore convex.  Thus, SGD will find a global optimum for the outer problem at each step.  In this way, convergence result for convex-concave minmax problems (e..g. [Wang & Li, 2020]) may apply, but one would need to prove this formally.
> > > > > > > > > > > > >
> > > > > > > > > > > > > * When $\Theta$ is compact and convex, the convexity of $f$ will mean that the canonical distribution defined in [Bubeck et al., 2015] will be log-concave, and therefore the speculative results from the last bullet point would still hold.
> > > > > > > > > > > > >
> > > > > > > > > > > > > * When $\Theta$ is not convex, the guarantees from [Bubeck et al., 2015] will not hold, because the distribution from which we want to sample in the inner problem will no longer be log-concave.  While there are preliminary results in this setting, e.g. [Raginsky et al., 2017], [Ma et al., 2019], [Zou et al., 2019], [Mangoubi et al., 2019], and [Ge et al., 2018], it remains to be seen how these analyses could be extended to Algorithm 1.
> > > > > > > > > > > > >
> > > > > > > > > > > > > This is the limit of what we know about convergence guarantees in this setting.  And we reiterate, these bullets provide **speculative ideas**.  While we plan to investigate this direction in a future manuscript, this is well beyond the scope of this paper, which -- as we have reiterated throughout these comments -- has several notable contributions, including: 1) lifting the well-known minimax robustness problem to a stochastic optimization problem over a specific, non-atomic perturbation distribution, (2) proposing an algorithm to (empirically) tackle this problem, (3) showing that our algorithm performs on-a-par with SOTA algorithms, and (4) providing PAC-style generalization results for our algorithm.
> > > > > > > > > > > > >
> > > > > > > > > > > > > ---
> > > > > > > > > > > > >
> > > > > > > > > > > > >
> > > > > > > > > > > > > ### References
> > > > > > > > > > > > >
> > > > > > > > > > > > >
> > > > > > > > > > > > > [Bubeck et al., 2015] Sebastien Bubeck, Ronen Eldan, and Joseph Lehec. "Finite-time analysis of projected Langevin Monte Carlo." Advances in Neural Information Processing Systems. 2015.
> > > > > > > > > > > > >
> > > > > > > > > > > > > [Wang & Li, 2020] Yuanhao Wang, and Jian Li. "Improved algorithms for convex-concave minimax optimization." arXiv preprint arXiv:2006.06359 (2020).
> > > > > > > > > > > > >
> > > > > > > > > > > > > [Raginsky et al., 2017] Maxim Raginsky, Alexander Rakhlin, Matus Telgarsky. Non-Convex Learning via Stochastic Gradient Langevin Dynamics: A Nonasymptotic Analysis. COLT, 2017.
> > > > > > > > > > > > >
> > > > > > > > > > > > > [Ma et al., 2019] Yi-An Ma, Yuansi Chen, Chi Jin, Nicolas Flammarion, Michael I. Jordan. Sampling can be faster than optimization, PNAS, 2019.
> > > > > > > > > > > > >
> > > > > > > > > > > > > [Zou et al., 2019] Difan Zou, Pan Xu, Quanquan Gu. Sampling from Non-Log-Concave Distributions via Stochastic Variance-Reduced Gradient Langevin Dynamics. AISTATS, 2019.
> > > > > > > > > > > > >
> > > > > > > > > > > > > [Mangoubi et al., 2019] Oren Mangoubi, Nisheeth K. Vishnoi. Nonconvex sampling with the Metropolis-adjusted Langevin algorithm. COLT, 2019.
> > > > > > > > > > > > >
> > > > > > > > > > > > > [Ge et al., 2018] Rong Ge, Holden Lee, Andrej Risteski. Simulated Tempering Langevin Monte Carlo II: An Improved Proof using Soft Markov Chain Decomposition. arXiv preprint arXiv:1812.00793 (2018).
> > > > > > > > > > > > >
> > > > > > > > > > > > > [Gao et al., 2019] Ruiqi Gao, Tianle Cai, Haochuan Li, Cho-Jui Hsieh, Liwei Wang, and Jason D. Lee. "Convergence of adversarial training in overparametrized neural networks." Advances in Neural Information Processing Systems 32 (2019): 13029-13040.

---

> > > > > > > > > > > > > > ### Comment · Reviewer_rhY5 · 2021-09-02
> > > > > > > > > > > > > > **Thanks for clarification**
> > > > > > > > > > > > > >
> > > > > > > > > > > > > > Thanks for the authors' response. The response clarifies my question about whether the proposed algorithm is guaranteed to converge or not.
> > > > > > > > > > > > > >
> > > > > > > > > > > > > > > If the reviewer stands by this claim, then we emphasize that no adversarial training algorithm possesses such guarantees.
> > > > > > > > > > > > > >
> > > > > > > > > > > > > > (**This is not a question, just clarify my previous comments.**) I attempted to clarify whether the proposed algorithm can be guaranteed to converge or not. This serves as a motivation for checking whether '**lifting the well-known minimax robustness problem to a stochastic optimization problem over a specific, non-atomic perturbation distribution**' could provide additional benefits (in terms of algorithms) compared with the original min-max formulation or not. Also, I need to clarify that theoretical guarantees are not limited to convergence guarantees. For example, previous works [1, 2, 3] provide certification guarantees for adversarial examples.
> > > > > > > > > > > > > >
> > > > > > > > > > > > > > [1]. Eric Wong and Zico Kolter. "Provable defenses against adversarial examples via the convex outer adversarial polytope." International Conference on Machine Learning. PMLR, 2018.
> > > > > > > > > > > > > >
> > > > > > > > > > > > > > [2]. Aditi Raghunathan, Jacob Steinhardt, and Percy Liang. "Certified defenses against adversarial examples." arXiv preprint arXiv:1801.09344 (2018).
> > > > > > > > > > > > > >
> > > > > > > > > > > > > > [3]. Jeremy Cohen, Elan Rosenfeld, and Zico Kolter. "Certified adversarial robustness via randomized smoothing." International Conference on Machine Learning. PMLR, 2019.

---

> > > > ### Comment · Reviewer_rhY5 · 2021-08-23
> > > > **Quick response**
> > > >
> > > > Thanks for the detailed response.
> > > >
> > > > >To summarize, based solely on the empirical results in [Rice et al., 2020], it is hard to unilaterally conclude that adversarially robust generalization is impossible, especially given that this is at odds with the theoretical results discussed in the previous section.
> > > >
> > > > Yes, I agree adversarially robust generalization is possible. However, my point is directly assuming this assumption (9a) holds is not reasonable.
> > > >
> > > > >If the reviewer is interested, we would be happy to provide further references in support of Assumption 3.5 holding in practice.
> > > >
> > > > It would be very helpful if the authors could provide the empirical support of Assumption 3.5. As far I as know, I do not know any empirical results suggest Assumption 3.5 holds for CIFAR-10 ($\ell_{\infty}=8/255$) in practice.

---

### Official Review · Reviewer_v1iK · 2021-07-17

**Rating:** 7
**Confidence:** 3

**Summary:**

The paper provided a novel connection between the well-known adversarial training problem and a statistical problem over perturbation distributions. The former requires obtaining worst-case adversarial perturbations of a neural network as an inner optimization procedure, which has been known as a challenging task. The later can avoid this but needs to solve a new inner optimization over perturbation distributions instead. The authors demonstrated that the statistical problem recovered several existing adversarial training problems with sub-optimal perturbation distributions. Inspired by this connection, the paper introduced an alternative adversarial training algorithm that used a smoothed version of the optimal perturbation distribution and showed that the proposed algorithm can achieve competitive or even better performance compared to previous approaches in terms of empirical adversarial robustness (evaluated using empirical attacks) on image classification tasks such as MNIST and CIFAR-10.

**Limitations And Societal Impact:**

I think that the paper can be benefited from adding more discussion of the limitations of the proposed algorithm. For example,
- What is the run-time of the proposed algorithm? (e.g., how large does $L$ need to be to be competitive of existing approaches and what is the resulting run-time?)
- The proposition 3.2 assumed that the loss function $(x, y) \mapsto \ell(f_\theta(x), y) \in L^2$. Does this assumption usually hold for the neural networks of interest?

**Main Review:**

The writing is clear and easy to follow. Even though there are some typos/missing definitions in the main body, I believe they are largely innocuous and do not hinder understanding.

My main concern of the paper at the current state is its lack of empirical evaluations:

1) *Lack of evaluations on the trade-off between using more Langevin iterations vs. model performance*
The proposed method uses projected Langevin sampler to approximate the optimal distribution of the perturbation.
Having enough number of steps (L) in the sampling step is usually crucial to the quality of the approximation, but can incur a big computation cost at the same time.
Because of this, I believe it is important to analyze the trade-off between $L$/training time and the performance, and I think the current paper does not do that thoroughly.
The authors did provide some results to show the difference in performance with different $L$'s. For example, the performance is reported with $L=1$ and $L=100$ for MNIST and $L=10$ for CIFAR-10, but they alone don't tell much about how practical these settings are when scaling up to larger networks without learning about their training time.

2) *Lack of experimental details*
The experimental section didn't specify the threat model (e.g., under which norm ball the perturbation should be confined in).
This is an important detail that should be included since the difficulty of robust classification varies greatly among different norms.
Also, the hyperparameters for the PGD attack evaluations are also missing.
As the authors also noted that adversarial attacks rely on "heuristics such as random initializations and random starts, ...", providing these details and the tuning procedures for the attacks would be helpful for a more thorough examination of the proposed method's performance.

Aside from these two main issues, the experiment section could also be benefited from a more diverse set of perturbation radius and perturbation shape (e.g., different norms).

*Some More Detailed Comments*
- Algorithm 1, L8 - What is the difference between $\tilde{l}$ and $l$? Is it a typo?
- Algorithm 1, L5 - I think $g, T$ should be defined somewhere in the main body rather than only in the Appendix.
- L238 - same as above. I think $\mathcal{H}$ is never defined in the main body.
- L303 - typo "it is well-know that..." => well-known
- The references are missing for some baselines in the experiment section (e.g., FAB, A-PGD, MART).

Overall, I think it is a really wonderful paper with a novel reformulation of the adversarial learning problem accompanied with an effective practical algorithm. I would recommend acceptance and I think improving the empirical section could greatly further strengthen the paper.

**Time Spent Reviewing:**

10

---

> ### Author Response · Authors · 2021-08-10
> **Sample complexity clarification + new L_2 experiments + larger sweep over perturbation radii**
>
> We thank the reviewer for their thoughtful review.  We are delighted they feel that "[this] is a really wonderful paper with a novel reformulation of the adversarial learning problem accompanied with an effective practical algorithm."  Below, we address some of the main points raised in their review.
>
> ## Runtime complexity
> The complexity of Algorithm 1 is essentially the same as iterative adversarial training techniques such as projected gradient methods (see Appendix A). The number of Langevin iterations L is equivalent to the number of steps in the inner loop of these approaches. For comparison, our CIFAR-10 benchmark uses L=10 iterations/steps for DALE, PGD, TRADES. The complexity of DALE is therefore not larger than currently established techniques.  The run time results shown in the table below for MNIST corroborate this observation.  We report runtimes in the number of seconds per batch (batch size = 64).
>
> | Num. Steps | PGD sec/batch | TRADES sec/batch| DALE sec/batch |
> | :-------: | :--------: | :------------: | :------------: |
> | 1 | 0.013 | 0.029 | 0.016 |
> | 5 | 0.053 |  0.047 | 0.063 |
> | 10 | 0.120 | 0.126 | 0.126 |
> | 30 | 0.353 | 0.380 | 0.381 |
>
> These details are indeed not clear in the current version of the manuscript and we intend to use the additional space in the camera-ready to include more precise descriptions of our experimental setup.
>
> ## New experiments
> As the reviewer suggested, we will also include experiments with different perturbation models. Preliminary results on MNIST (shown in the table below) show that for l2-bounded perturbations (ε = 1.0, 1.5, and 2.0), the clean and adversarial accuracy of DALE remains as good or better than PGD-l2 (each row is formatted as "clean accuracy/adversarial accuracy"):
>
> | Algorithm | Acc. ε=1.0 | Acc. ε=1.5 | Acc. ε=2.0 |
> | :-------: | :--------: | :--------: | :--------: |
> | PGD (l2) | 99.1/95.9 | 98.7/90.2 | 98.6/82.7 |
> | LMC-10 (l2) | 99.2/95.8 | 99.1/94.2 | 99.1/85.9 |
>
> We intend to add more baselines to this table in the camera-ready version as it highlights the flexibility of our method.
>
> We also show below some preliminary results for different radii with respect to l_inf bounded attacks on MNIST. Note that DALE is on a par with the baselines and in some cases improves by up to two percentage points (each row is formatted as "clean accuracy/adversarial accuracy").
>
> | Algorithm | Acc. ε=0.1 | Acc. ε=0.2 | Acc. ε=0.3 |
> | :-------: | :--------: | :--------: | :--------: |
> | PGD | 99.2/97.7| 99.1/96.1 | 98.1/93.1|
> | TRADES | 99.1/97.3| 99.0/96.2 | 98.9/94.0 |
> | LMC-5 | 99.5/98.0 | 99.4/96.0 | 99.1/92.6 |
> | LMC-10 | 99.4/98.1 | 99.3/96.2 | 99.0/92.9 |
> | LMC-30 | 99.3/97.9 | 99.3/96.4 | 99.1/95.0 |
>
> ## L2 assumption
> The reviewer has a point that we never discussed the consequences of this assumption in the text. As it turns out, this hypothesis, though necessary, is mostly technical and mild in practice. Indeed, if the loss is bounded (a common assumption in learning theory), then square integrability obtains immediately regardless of the model. Since the loss and model are typically continuous, then it is enough, e.g., for the feature/target space Omega = (X,Y) to be compact (bounded and closed). Note that the loss/model map must be integrable (L1) for the expectations to exist, so requiring that they be in L2 is only slightly more restrictive. We will make this clearer in the camera-ready version of the manuscript.
>
> ## Definitions and typos
> The reviewer is correct that in [Algorithm 1, L8] the loss $\tilde{l}$ should be $\bar{l}$ described in the text (line 281). We will copy-edit the manuscript to make sure that all symbols are defined in the main text, particularly those listed here.

---

### Official Review · Reviewer_JwwU · 2021-08-03

**Rating:** 6
**Confidence:** 5

**Summary:**

The paper presents a new algorithm for performing adversarially robust training via a dual characterization.

**Limitations And Societal Impact:**

Some technical limitations of the work are addressed.

**Main Review:**

This paper considers the problem of learning adversarially robust classifiers. The problem is traditionally formulated as that of minimizing the robust loss where on any given point (x,y) the loss is evaluated at (x+delta, y) where delta is a perturbation among a set of allowed perturbations. The loss at (x,y), \ell(x,y) is then the maximum loss in the neighborhood of x, i.e, \ell((x+\delta, y)). The robust loss is intractable in general and in fact even the maximization problem of finding the worst perturbation for a given (x,y) is hard unless the classifiers is a simple model, such as linear classifiers.

The paper proposes to solve the constrained robust learning problem where the goal is to minimize the robust loss subject to the fact that the standard (non-robust) loss is small as well. This formulation addresses practical concerns where optimizing for the robust loss typically leads to an increase in the standard loss. The authors propose a dual formulation of the problem by providing a dual characterization of the inner maximization problem. In the dual space the inner maximization can be written as optimizing over a set of distributions over perturbations (for a given (x,y)). The main result of the paper is a characterization of the optimal distribution. Based on this the authors propose a sampling approach, where for each (x,y) the perturbation is sampled from an approximation to the optimal perturbation distribution. The approximation essentially samples a perturbation delta with probability proportional to \ell(x+delta, y). The authors present empirical results on CIFAR-10 and MNIST and show that the method proposed is competitive with existing approaches.

Overall the paper presents a novel take on adversarial robustness. Proposition 3.2 is the key result. It would be interesting to consider better approximations to the optimal distribution than the choice made in the paper (given that the empirical improvements obtained are within 1-2%).
The paper also presents generalization bounds, but they more of less follow standard techniques.


**Time Spent Reviewing:**

3

---

> ### Author Response · Authors · 2021-08-10
> **Further clarifications concerning approximations of the optimal perturbation distribution & generalization bounds**
>
> We thank the reviewer for their thoughtful review.  We are glad they feel the "paper presents a novel take on adversarial robustness."
>
> With regard to the comment:
>
> > " It would be interesting to consider better approximations to the optimal distribution than the choice made in the paper."
>
> We consider only an approximation of the optimal perturbation distribution in our algorithm due to the difficulty of sampling from discontinuous distributions in high dimensions.  While this approximation already leads to good results (in some cases, even improving on the state-of-the-art), we do expect better performance to come from improving the sampling step of our algorithm (e.g., using recent advances in discontinuous HMC such as [88]). As the reviewer noted, however, this paper focused on the developing the theoretical foundations of this approach, leaving these algorithmic improvements for future work.
>
> While we agree with the reviewer that our generalization results are based on standard techniques, they provide fruitful insight on the impact of using constrained learning as opposed to regularization approaches. We have also begun exploring how the new problem characterization in Propositions 3.1--3.2 may provide a different avenue to tackle the issue of adversarial generalization.

---

### Decision · Program_Chairs · 2021-09-27

**Decision:**

Accept (Poster)

**Comment:**

The paper proposes to solve the constrained robust learning problem via a dual formulation of the problem by where the inner maximization can be written as optimizing over a set of distributions over perturbations (for a given (x,y)). The main result of the paper is a characterization of the optimal distribution. Based on this the authors propose a sampling approach, where for each (x,y) the perturbation is sampled from an approximation to the optimal perturbation distribution. The paper presents a practical algorithm based on the theory that achieves competitive results on benchmark datasets.

The reviewers raised concern about some of the theoretical portions of the paper that follow from standard tools. Given that the empirical results are competitive with algorithms such as TRADES (that are also based on theoretical insights), it was unclear whether the main theoretical result (proposition 3.2) is novel enough for NeurIPS. It was also pointed out by the reviewers that the empirical comparisons where not conducted with respect to proper benchmarks such as RobustBench (this was addressed by the authors in the response via additional experimental results). Overall this is a borderline paper.